# SPECIALIZED FOUNDATION MODELS STRUGGLE TO BEAT SUPERVISED BASELINES

**Zongzhe Xu,**[*] **Ritvik Gupta,**[*] **Wenduo Cheng, Alexander Shen, Junhong Shen**
Carnegie Mellon University
`{zongzhex,ritvikgu,wenduoc,ajshen,junhongs}@andrew.cmu.edu`
[*] denotes equal contribution; order decided by coin flip

**Ameet Talwalkar**
Carnegie Mellon University & Datadog, Inc.
`talwalkar@cmu.edu`

**Mikhail Khodak**
Princeton University
`mkhodak@cs.princeton.edu`

## ABSTRACT

Following its success for vision and text, the "foundation model" (FM) paradigm—pretraining large models on massive data, then fine-tuning on target tasks—has rapidly expanded to domains in the sciences, engineering, healthcare, and beyond. Has this achieved what the original FMs accomplished, i.e. the supplanting of traditional supervised learning in their domains? To answer we look at three modalities—genomics, satellite imaging, and time series—with multiple recent FMs and compare them to a standard supervised learning workflow: model development, hyperparameter tuning, and training, all using only data from the target task. Across these three specialized domains, we find that it is consistently possible to train simple supervised models—no more complicated than a lightly modified wide ResNet or UNet—that match or even outperform the latest foundation models. Our work demonstrates that the benefits of large-scale pretraining have yet to be realized in many specialized areas, reinforces the need to compare new FMs to strong, well-tuned baselines, and introduces two new, easy-to-use, open-source, and automated workflows for doing so.

## 1 INTRODUCTION

Recent years have witnessed a shift towards large-scale pretraining across domains like computer vision and natural language processing. This workflow generally consists of two stages: pretraining on vast amounts of domain-specific data to capture general knowledge followed by fine-tuning on target tasks (Radford & Narasimhan, 2018). This pretrain-then-finetune paradigm has been tremendously successful, enabling foundation models (Bommasani et al., 2021) to consistently outcompete traditional supervised learning methods on a wide variety of downstream tasks in the vision and language domains (Dosovitskiy et al., 2021; Liu et al., 2021; Devlin et al., 2019).

Driven by this success, the foundation model approach has been adapted to various *specialized* domains, which we define to be ML application areas—e.g. genomics, satellite imaging, and time series—whose data modalities lie outside those of classical AI tasks, i.e. natural images and text. Such domains have seen the introduction of many new FMs claiming to leverage large pretraining datasets to achieve breakthrough performance on downstream tasks (Dalla-Torre et al., 2023; Nguyen et al., 2024; Zhou et al., 2023b; Avsec et al., 2021; Ji et al., 2021; Fuller et al., 2023; Cong et al., 2022; Mendieta et al., 2023). These claims underlie our study's motivating question:

*Do these new specialized FMs outperform traditional supervised learning applied to the same tasks?*

Answering this question is critical because supervised workflows are usually much less expensive to implement and deploy, but FMs that allow for effective transfer learning have the potential to fundamentally transform these domains, as we have seen with language and vision processing in the past decade. However, despite ongoing efforts to promote their fair and comprehensive evaluation (Liang et al., 2022; Bommasani & Liang, 2021), many new FMs have not been adequately compared to simpler, often more efficient baselines. Indeed, we found that many works only benchmark their proposed models against other FMs, essentially creating a comparison echo chamber (Fuller et al., 2023; Mendieta et al., 2023; Nguyen et al., 2024; Zhou et al., 2023b).

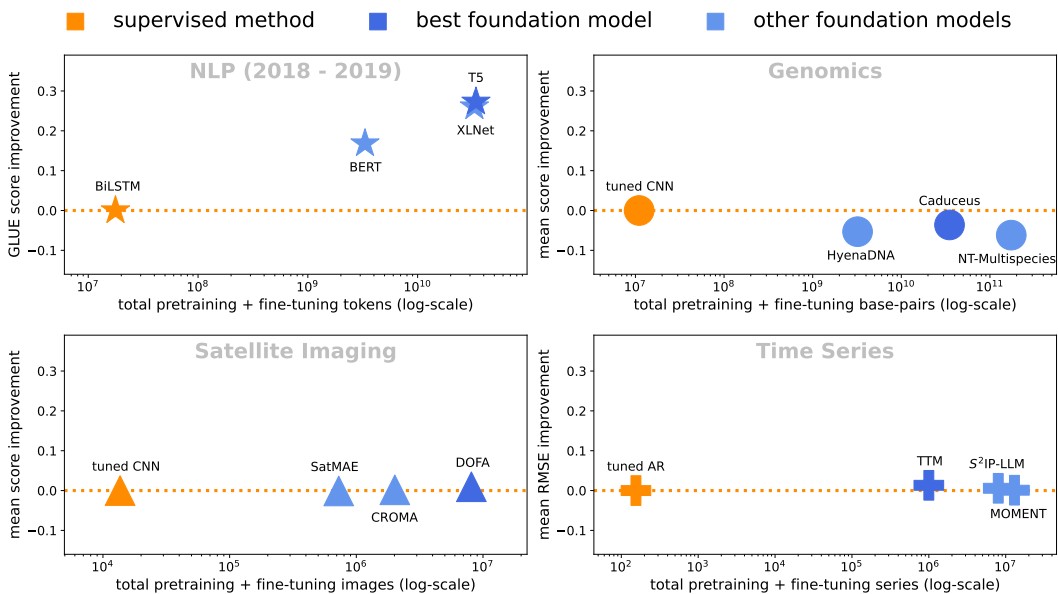

Figure 1: Across genomics, satellite imaging, and time series, specialized FMs fail to significantly improve upon supervised learning despite using two-to-five orders of magnitude more data. In contrast, breakthrough FMs such as BERT dramatically outperformed supervised NLP baselines (top left), causing the field to switch to fine-tuning as the default approach. For each domain we plot total pretraining and fine-tuning data vs. mean improvement over the supervised state-of-the-art. The NLP results are from the GLUE benchmark (Wang et al., 2019) while evaluations of the last three domains are in Section 4. Note that for the NLP x-axis we ignore word embedding pretraining tokens.

We answer our motivating question by considering a representative set of three specialized domains—chosen according to the presence of multiple FMs and a standard set of evaluation tasks—and comparing their performance on those tasks with that of a traditional supervised learning workflow. The latter is a three step process (i.e., model development, hyperparameter tuning, training) in which all steps use only data from the target task, in contrast to the FM workflow, which uses vast amounts of pretraining data (c.f. Figure 2). By leveraging model selection tools ranging from classical information criteria to cutting-edge architecture search, we build automated pipelines that efficiently develop and train strong supervised models on over fifty tasks across three distinct domains.

Our main result is negative: despite pretraining on massive data, specialized FMs struggle and often fail to outperform models trained solely on downstream task data with traditional supervised learning (c.f. Figure 1). Specifically, lightly adapted convolutional neural network (CNN) architectures such as wide ResNet and UNet attain state-of-the-art on the Nucleotide Transformer benchmark in genomics and match the latest satellite FMs on downstream classification. Furthermore, tuned linear auto-regression (AR) matches or outperforms every open-source time series FM on a standard suite of forecasting tasks, despite using four or more orders of magnitude fewer parameters and data.

These findings demonstrate that genomics, satellite imaging, and time series have not yet had their "BERT moment" (Devlin et al., 2019), i.e. these domains have not yet pretrained FMs that dominate traditional supervised approaches. This is despite the fact that all of them have BERT-scale[1] FMs and many are already witnessing a shift towards not comparing with supervised approaches, as was seen in natural language processing (NLP) post-BERT. More broadly, since these domains are among the most high-profile areas with specialized FMs, our results suggest that practitioners should still consider supervised models in addition to FMs. They also reinforce the need for robust and well-tuned baselines, with surprising findings such as (a) simply tuning kernel sizes and dilation rates in standard CNNs dominates a genomics benchmark and (b) rescuing the century-old AR forecaster from obsolescence is as easy as considering lookback parameters larger than five and training on a GPU. To facilitate ongoing research in these and other domains, we make code associated with both our CNN-tuning pipeline (DASHA) and our AR-on-GPU workflow (Auto-AR) publicly available.[2]

---

[1] Models with 100M+ parameters trained on 100x or more data than supervised tasks in the domain are given.

[2] Available at https://github.com/ritvikgupta199/DASHA and https://github.com/Zongzhe-Xu/AutoAR.

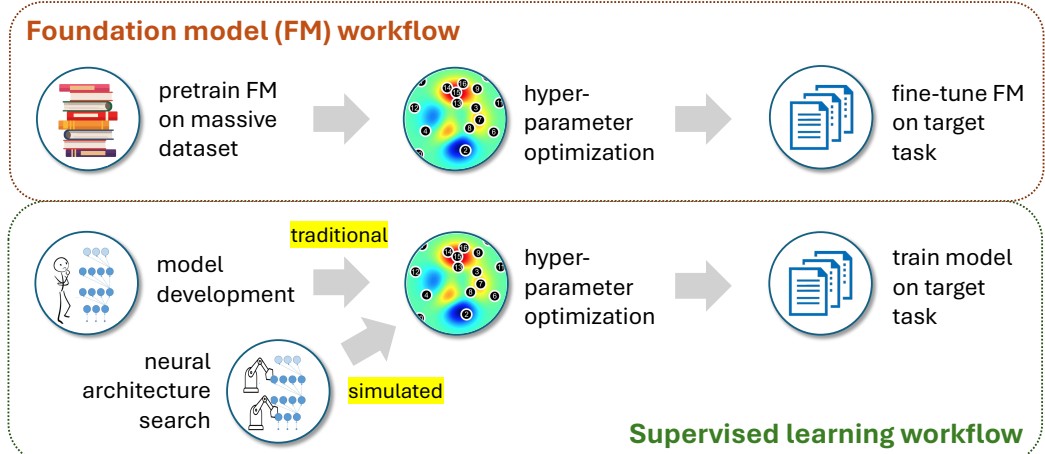

Figure 2: Our goal is to compare the pretrain-then-fine-tune paradigm (top) with a standard supervised workflow (bottom) on the tasks on which specialized FMs are evaluated. While for time series we go through a traditional process of developing and tuning a supervised model, this manual approach does not scale to many domains; as a result, in Section 3.1 we develop a way to simulate it using architecture search. Note that FM fine-tuning hyperparameters are not always tuned in practice, but we assume their creators make a best-effort attempt to present their own method in the best light.

## 2 RELATED WORK

Foundation models have been trained in numerous specialized domains beyond vision and text, including genomics (Ji et al., 2021), satellite imaging (Cong et al., 2022), time series (Goswami et al., 2024), weather (Bodnar et al., 2024), pathology (Zimmermann et al., 2024), differential equation solving (Sun et al., 2024), web traffic (Zhao et al., 2023), and beyond. To get a representative sense of their success, we focus on domains that combine the following properties: (a) multiple BERT-scale FMs, (b) a standard suite of evaluation tasks, and (c) significant applied interest. These restrictions suggest looking at three domains, all of which have at least five FMs evaluated on at least nine tasks: genomics (which has some of the largest-available non-text FMs (Dalla-Torre et al., 2023)), satellite imaging (which has a large ongoing benchmarking effort (Lacoste et al., 2024)), and time series (which has already seen significant industry interest (Cohen et al., 2024)). The remainder of this section examines how different learning workflows approach problems in these domains.

**Specialized foundation models:** Collectively our three target domains have more than thirty FMs, many developed via the "lift-and-shift" approach—borrowing terminology from Rolf et al. (2024)—in which techniques from core AI areas such as vision and language processing are applied with modest tailoring to specialized domains. In particular, many methods are built on out-of-domain models such as BERT, Swin, and Hyena (Ji et al., 2021; Mendieta et al., 2023; Nguyen et al., 2024; Shen et al., 2024b), with adaptations including specialized tokenizations, embeddings, and model modifications for handling domain-specific considerations like long-range dependencies (Dalla-Torre et al., 2023; Zhou et al., 2023b; Shen & Yang, 2021; Das et al., 2023; Shen et al., 2024a;c; 2025; Cohen et al., 2024; Liang et al., 2025) or multispectral data (Cong et al., 2022). While the "lift-and-shift" approach can often be useful or at least a good starting point, its widespread use underlines the need for strong *in-domain* baselines to make sure that the combination of out-of-domain tooling and massive pretraining data is actually helpful. Such comparisons are not always conducted, e.g. the satellite FM SatMAE (Cong et al., 2022) is compared to ImageNet-initialized and randomly initialized ResNet-50 (He et al., 2015), while most of the time series FMs we consider only do a full comparison to one linear baseline, DLinear (Zeng et al., 2023). This can sometimes be justified—e.g. in the case of NLP post-BERT—but our results suggest that for now, specialized FMs should still compare to in-domain supervised model development.

We note that we are not the first to take a critical look at specialized FMs. For example, Yang et al. (2024) questioned the dominance of Transformers for protein sequence FMs by showing that convolutions could do just as well, which is related to our discovery that (supervised) CNNs were competitive with (largely Transformer-based) genomics FMs. Another study by Kedzierska et al. (2023) found that training an in-domain generative model could outperform pretraining in single-cell biol-

ogy applications. In satellite imaging, Jakubik et al. (2023) showed that a supervised UNet performs competitively with an FM on a single non-standard task, while Xiong et al. (2024) found that supervised models sometimes outperform linearly probed FMs. Corley et al. (2024) further demonstrated that ImageNet-pretraining can be effective in this domain when properly tailored. Finally, in the time series domain, Tan et al. (2024) show that the popular approach of fine-tuning text-pretrained LLMs on time series tasks often underperforms supervised models (in their case randomly initialized attention); our results generalize this to other time series FMs and use an even simpler supervised model (AR) to do so. Overall, these studies focus on specific cases, whereas our results identify a broader trend of FM underperformance and/or weak baselines across multiple fields.

**Specialized baselines:** Both automated supervised pipelines that we develop are heavily influenced by successful in-domain model development. In particular, the NAS-based pipeline we use to achieve our results in genomics and satellite imaging is inspired by the success of the human-driven specification of kernel sizes and dilation rates in successful architectures like TCN (Lea et al., 2016) and ConvNeXt (Liu et al., 2022). At the same time, for time series our approach is based upon a well-tuned GPU implementation of perhaps the most basic forecasting model, AR.

**AutoML for specialized domains:** While often evaluated on domains such as vision, automated techniques are also important in specialized domains. An important example is Auto-ARIMA (Hyndman & Khandakar, 2008) for time series, although it underperforms on the tasks we consider (Challu et al., 2022). However, to avoid requiring significant expertise in any one domain, we also make use of AutoML methods developed specifically for diverse tasks (Roberts et al., 2021b; Shen et al., 2023), in particular the NAS method DASH (Shen et al., 2022) that can discover good kernel sizes and dilation rates for a CNN backbone faster than it can be trained from scratch.

## 3 METHODOLOGY

Recall that our goal is to conduct a robust comparison between traditional supervised learning and specialized FMs; the natural way to do this is to take existing benchmarks used to evaluate FMs in our three target domains and run a typical supervised workflow on the same tasks. As depicted in Figure 2, this pipeline involves three steps: (1) model development, (2) hyperparameter tuning, and (3) training. The first stage involves using both reasoning and trial-and-error to find a good architecture to tune and train on the data; for example, Lea et al. (2016) developed the temporal convolutional network (TCN) architecture with a multi-layer dilation rate pattern specifically suited to sequential data, while Liu et al. (2022) designed the breakthrough ConvNeXt architecture by methodically exploring ways to make CNNs more like Transformers without introducing attention. The second stage (hyperparameter tuning) can also be done via human-driven iteration, but there exist effective automated procedures for it as well (Li et al., 2020). Lastly, the third step of the pipeline involves simply training the selected model with the selected configuration on the data of the target task.

While it is standard to automate the last two steps of the procedure, model development is typically done by hand and so is difficult to do for fifty tasks across three domains. As a result, we settle for *approximating* the traditional supervised workflow by simulating the model development component via neural architecture search. To ensure fair comparison and reduce computation, we use low-fidelity NAS methods that return an architecture in less time than it takes to train it. Our results can thus be viewed as *lower bounds* on the performance of supervised learning, as model development might be significantly improved using less-heuristic and/or human-driven architecture design.

In the remainder of this section we detail how we handle the different steps of the supervised pipeline. Note that our NAS-dependent workflow (DASHA)—which we cover in the first part of this section—yields our main results for genomics and satellite imaging but *not* for time series. There we find its performance to be less competitive and instead focus on an even simpler approach based on linear auto-regression, whose model development and tuning we describe in the second subsection.

### 3.1 DASHA: SIMULATING THE SUPERVISED WORKFLOW USING NAS

To simulate model development we need a search space over architectures that is (a) efficient, (b) flexible, and (c) applicable to the types of high-dimensional unstructured data that arise in domains targeted by specialized FMs; these requirements make CNN-based search spaces a natural choice. In particular, inspired by the success of hand-tuned kernel sizes and dilation rates in traditional model development (Lea et al., 2016; Bai et al., 2018; Liu et al., 2022), we apply DASH (Shen et al., 2022), a NAS method that starts with an existing CNN backbone—e.g. a wide ResNet (Zagoruyko & Komodakis, 2017)—and uses the weight-sharing heuristic (Liu et al., 2018)

---

**Algorithm 1:** Pseudocode for the DASHA workflow. Starting with a set of backbone CNNs, we use DASH (Shen et al., 2022) to set the right kernel size and dilation rate for each of its convolutional layers and then use ASHA (Li et al., 2020) to configure a training routine for the resulting architecture. Lastly, we pick the best backbone using validation data and train it.

---

**Input:** target task dataset $D$, candidate CNN backbone architectures $A$
**for** *CNN backbone $a \in A$* **do**
  // set a kernel size and dilation rate for each layer of $a$
  $\mathrm{arch}_a \leftarrow \mathrm{DASH}(D, a)$
  // tune hyperparameters for the discovered architecture $\mathrm{arch}_a$
  $\mathrm{config}_a, \mathrm{val\_score}_a \leftarrow \mathrm{ASHA}(D, \mathrm{arch}_a)$
// train the architecture with the highest validation score
$a \leftarrow \arg\max_{a \in A} \mathrm{val\_score}_a$
**Output:** $\mathrm{train}(D, \mathrm{arch}_a, \mathrm{config}_a)$

---

to determine the right kernel size and dilation rate to use at each convolutional layer. DASH has been successfully used in AutoML competitions (Roberts et al., 2021a) and to advance the state-of-the-art on NAS benchmarks (Tu et al., 2022), making it likely to be useful beyond the domains we consider.

As described in Algorithm 1, we augment DASH in two ways: (1) trying more than one CNN backbone (e.g. both wide ResNet and UNet (Ronneberger et al., 2015)) and (2) using the popular hyperparameter optimization (HPO) method ASHA (Li et al., 2020) to configure training settings for the discovered model. Following the NAS and tuning stages, we train the discovered architecture with the selected configuration on the target data. Further details, including the resources given to the three steps of the pipeline and the exact search spaces used by DASH and ASHA, are provided in Appendix B.1. Note that, while our focus is on *data*-efficient baselines, we do ensure that the entire workflow is never substantially more computationally expensive than fine-tuning an FM.

### 3.2 AUTO-AR: MAKING A BASELINE STRONGER BY MAKING IT SIMPLER

While DASHA can be applied to forecasting tasks, it is not competitive with state-of-the-art time series FMs. At the same time, the field of time series forecasting has long employed automated workflows, notably the Auto-ARIMA approach of Hyndman & Khandakar (2008) that uses statistical tests and information criteria to tune ARIMA's lookback and differencing parameters. Auto-ARIMA was evaluated on the time series tasks we consider by Challu et al. (2022), who found that it performed poorly compared to deep learning approaches. However, their implementation does not make use of multi-channel data and tunes up to a lookback window of at most five, which is much less data than used by time series FMs. While tuning ARIMA with larger lookback parameters is computationally costly, we find the following simplified tuning pipeline to be effective:

1. use the KPSS test (Kwiatkowski et al., 1992) to decide whether to take first differences
2. use the Bayesian Information Criterion to select the maximum lookback parameter of the auto-regressive (AR) component of ARIMA, ignoring the moving average (MA) part
3. maximize the multi-channel likelihood of AR with the chosen differencing and lookback

By dropping the MA component of the model and running the procedure on GPU, we are able to tune the lookback windows up to the maximum allowable length (usually 512); we find that longer lookbacks are critical for performance. Note that this is just a tuned version of the classic AR model.

## 4 EMPIRICAL RESULTS

We now present the results of applying the automated pipelines described in the previous section to our three target domains. For each domain, we provide a brief justification of the specific FMs and evaluation tasks that we consider, followed by details on how we apply our workflows; further information can be found in Appendices A and B. As there are too many separate results to present outside the appendix, in this section we mainly present aggregate statistics that summarize our findings for each domain, with detailed results relegated to Appendix C. The domains have different performance metrics, but they can all be aggregated via the following quantities: **average score**, **average rank**, and **mean / median percentage improvement over a baseline**. For each domain, we define a domain-specific baseline and measure the improvement of FMs and our approach relative to it. This standardizes comparisons across tasks of varying scales.

| | Model | Model Size | Pretraining Base-Pairs | Average Score ↑ | Average Rank ↓ | Mean %Imp.↑ | Median %Imp.↑ |
|---|---|---|---|---|---|---|---|
| **Foundation Models** | Enformer | 252M | 4B | 0.569 | 11.86 | 27.73 | 27.91 |
| | NT-1000G (500M) | 500M | 20.5T | 0.625 | 10.52 | 33.48 | 36.74 |
| | NT-1000G (2.5B) | 2.5B | 20.5T | 0.656 | 7.0 | 36.58 | 40.86 |
| | NT-Multispecies (500M) | 500M | 174B | 0.700 | 3.81 | 40.76 | 45.07 |
| | NT-Multispecies (2.5B) | 2.5B | 174B | 0.697 | 4.08 | 40.51 | 45.52 |
| | DNABERT-2 | 117M | 32.5B | 0.680 | 6.88 | 38.65 | 43.59 |
| | HyenaDNA-1K | 1.6M | 3.2B | 0.708 | 6.92 | 41.2 | 43.36 |
| | HyenaDNA-32K | 1.6M | 3.2B | 0.630 | 10.22 | 33.96 | 36.93 |
| | Caduceus-PS | 1.9M | 35B | 0.689 | 6.69 | 39.08 | 41.38 |
| | Caduceus-PH | 1.9M | 35B | 0.725 | 4.69 | 42.63 | 45.01 |
| **Supervised Models** | Wide ResNet | 2.0M | 0 | 0.694 | 6.83 | 37.16 | 43.08 |
| | UNet | 4.5M | 0 | 0.68 | 7.78 | 38.67 | 42.69 |
| | **DASHA (our workflow)** | 10.5M | 0 | **0.761** | **3.69** | **46.33** | **49.08** |

Table 1: Aggregate genomics performance, showing that our supervised workflow (DASHA) attains state-of-the-art on the NT benchmark, outperforming all FMs according to all measures while using no pretraining data and oftentimes many fewer parameters. For Mean / Median %Imp., we report % improvement over the Raw Probe baseline from Dalla-Torre et al. (2023), and for DASHA the model size refers to the largest configuration across tasks..

## 4.1 GENOMICS

We begin our investigation in the genomics domain, which has witnessed the development of numerous FMs, including the early Enformer (Avsec et al., 2021), the DNABERT series (Ji et al., 2021; Zhou et al., 2023b), the HyenaDNA family (Nguyen et al., 2024), GENA-LM[3] (Fishman et al., 2024), the recent Caduceus family (Schiff et al., 2024), and the NT family (Dalla-Torre et al., 2023); The latter includes models with up to 2.5B parameters. To evaluate them, we consider the Nucleotide Transformer (NT) benchmark of Dalla-Torre et al. (2023), which contains eighteen tasks in three main categories: regulatory elements, RNA production, and histone modification. We use this benchmark because of its diversity and because it has been evaluated on by all of the aforementioned FMs, allowing us to include eight of them in the comparison. Our numbers for these models are taken from Dalla-Torre et al. (2023, Supplementary Table 6); Following Dalla-Torre et al. (2023, Supplementary Table 5), We use F1 score and accuracy to evaluate a subset of regulatory elements and RNA production tasks, and we use Matthew's Correlation Coefficient (MCC) as the main metric for evaluation on the remaining datasets.

**Baselines:** CNNs have long been used for genomics tasks (Avsec et al., 2020; Cheng et al., 2024) and so are natural supervised baselines; in particular we include 1D variants of Wide ResNet (WRN) and UNet, which we find perform better than some domain-specific CNNs. We use these same two backbones as the candidate CNNs tuned and selected from by our DASHA workflow.

**Results:** Our genomics results are displayed in Table 1, which shows that our supervised workflow (DASHA) consistently outperforms all FMs across all aggregate metrics. As discussed in Appendix C, our strong performance is driven in large part by outstanding performance on the histone modification tasks (c.f. Table 9). The more detailed results also highlight the importance of considering diverse baselines, with Wide ResNet usually being the selected architecture but UNet performing significantly better for promoter and splice site classification tasks. Overall, DASHA sets a new state-of-the-art on the NT benchmark and certainly demonstrates that supervised methods remain quite competitive in genomics, despite the availability of massive pretraining datasets.

## 4.2 SATELLITE IMAGING

While they do not get as large as those in genomics, numerous BERT-scale FMs have also been introduced for satellite imaging, including SeCo (Manas et al., 2021), SatMAE (Cong et al., 2022), CROMA (Fuller et al., 2023), GFM (Mendieta et al., 2023), Scale-MAE (Reed et al., 2023), Satlas (Bastani et al., 2023), Prithvi (Jakubik et al., 2023), DOFA (Xiong et al., 2024), Clay (Clay Foundation, 2023), and SkySense (Guo et al., 2024). Because our evaluation includes GeoBench (Lacoste et al., 2024), a recently introduced satellite benchmark that has not been considered by many of these FMs, we obtain all results using our own fine-tuning; therefore we only consider a subset of top-performing open-source models. In all cases we use the fine-tuning workflow suggested by the

---

[3]We compare to GENA-LM in Appendix C, as it reports metrics that differ from those of the NT benchmark.

| | Model | Model Size | Pretraining Images | Average Score ↑ | Average Rank ↓ | Mean %Imp.↑ | Median %Imp.↑ |
|---|---|---|---|---|---|---|---|
| **Foundation Models** | SatMAE-Base | 85.6M | 700K | 76.99 | 9.89 | 5.27 | 3.59 |
| | SatMAE-Large | 303M | 700K | 77.75 | 7.56 | 6.52 | 4.62 |
| | GFM | 86.8M | 1.3M | 77.18 | 9.00 | 5.77 | 4.08 |
| | SwinT-Base | 86.8M | 14M | 76.69 | 8.72 | 4.86 | 1.43 |
| | CROMA-Base | 90.6M | 2M | 77.39 | 7.22 | 5.85 | 4.22 |
| | CROMA-Large | 312M | 2M | 78.03 | 5.44 | 6.90 | 6.09 |
| | DOFA-Base | 111M | 8M | 78.44 | 4.50 | 7.71 | 6.23 |
| | DOFA-Large | 337M | 8M | **78.80** | **3.50** | **8.40** | **6.88** |
| | Satlas | 90M | 45M | 77.47 | 6.33 | 5.87 | 3.86 |
| | ScaleMAE | 323M | 363.6K | 77.10 | 8.21 | 5.42 | 4.40 |
| | Clay | 92M | 70M | 77.33 | 7.50 | 5.99 | 3.51 |
| **Supervised Models** | ResNet50 | 23.5M | 0 | 73.76 | 13.22 | 0.30 | 00.07 |
| | Wide ResNet | 17.2M | 0 | 73.97 | 12.89 | 0.00 | 0.00 |
| | UNet | 17.3M | 0 | 75.73 | 9.94 | 3.01 | 1.07 |
| | **DASHA (our workflow)** | 32.4M | 0 | 77.85 | 6.06 | 6.67 | 5.16 |

Table 2: Performance on satellite imaging, demonstrating that a supervised workflow (DASHA) can get within 1% average accuracy of state-of-the-art specialized FMs while using no pretraining data and having 2-10× fewer parameters. For Mean / Median %Imp. we report % improvement over a vanilla Wide ResNet, and for DASHA the model size refers to the largest configuration across tasks.

authors of each FM plus some automated hyperparameter tuning; note that even with the extra tuning our reproductions on previous benchmarks systematically underperformed results reported in the original works. We select tasks mainly from GeoBench's five classification tasks and then add four additional tasks—BigEarthNet (Sumbul et al., 2019), EuroSAT (Helber et al., 2019), Canadian Cropland (Jacques et al., 2023), and fMoW-Sentinel (Cong et al., 2022)—that are commonly used to evaluate other FMs.[4] As we focus on classification, we report top-1 accuracy or mAP as appropriate.

**Baselines:** Since satellite imaging resembles RGB imaging, it is common to "lift-and-shift" vision models to this domain (Rolf et al., 2024). As a result we use several CNN backbones as baselines and wide ResNet as the candidate architecture for our DASHA workflow. Lastly, we also consider the performance of fine-tuning the ImageNet-pretrained vision FM SwinT-base (Liu et al., 2021).

**Results:** Table 2 shows that our supervised workflow attains competitive performance across all aggregate metrics and is only slightly outperformed by CROMA-large and the DOFA family. Unlike in genomics, the FMs here consistently outperform CNN backbones, likely because the associated papers compare to them as baselines. However, the frequently superior performance of DASHA suggests that domain-aware model development would yield good supervised models in this field. Another contrast with genomics is that the larger FM versions consistently attain superior performance here, suggesting they are making at least somewhat effective use of the pretraining data. Nevertheless, that this improvement can also be attained by DASHA, which uses no pretraining and produces a model that is 3-10× smaller, suggests that there remains significant room for improvement.

### 4.3 TIME SERIES

Our last domain is time series, which has many FMs that use the pretrain-then-fine-tune workflow, such as GPT4TS (OFA) (Zhou et al., 2023a), LLM4TS (Chang et al., 2023), MOMENT (Goswami et al., 2024), TEST (Sun et al., 2023), S$^2$IP-LLM (Pan et al., 2024), CALF (Liu et al., 2024), TTM (Ekambaram et al., 2024), and Time-LLM (Jin et al., 2024), and others that evaluate in a zero-shot (ZS) regime, such as TEMPO (Cao et al., 2024), TimesFM (Das et al., 2024), Moirai (Woo et al., 2024), and Toto (Cohen et al., 2024). We do not include four of these FMs in our main analysis (Table 16): for Time-LLM, this is because their results have been hard-to-reproduce for both us and past efforts (c.f. Appendix C.3.1), while the three others—Moirai, LLM4TS, and Toto—either evaluate on only a subset of our chosen tasks or are closed-source (or both); we list their reported numbers in Table 14. As discussed in Appendix C.3.2, Moirai underperforms Auto-AR while LLM4TS performs roughly on par with TTM (A). The most recent FM, Toto, has strong aggregate metrics, mainly due to a dominant performance on a single task, ETTh2 (c.f. Figure 7). As a result we do not view these excluded FMs as significantly affecting our conclusions. We also did not include another FM-like approach, TOTEM (Talukder et al., 2024), because of its focus on multi-task training rather

---

[4]In Appendix C we report results when excluding tasks where missing channels may affect performance.

| | Model | Model Size | Pretraining Series | Average RMSE ↓ | Average Rank ↓ | Mean %Imp.↑ | Median %Imp.↑ |
|---|---|---|---|---|---|---|---|
| **Foundation Models** | GPT4TS (OFA) | 87M | 8M | 0.555 | 6.70 | 31.29 | 22.71 |
| | TEST (Few Shot) | 345M | 8M | 0.603 | 10.70 | 25.23 | 14.59 |
| | MOMENT | 385M | 13M | 0.550 | 5.14 | 31.95 | 23.14 |
| | TTM (B) | 1M | 1M | 0.543 | 2.89 | 32.92 | **25.36** |
| | TTM (A) | 5M | 1M | **0.538** | **2.21** | **33.38** | 24.96 |
| | S$^2$IP-LLM | 345M | 8M | 0.545 | 3.54 | 32.59 | 24.63 |
| | CALF | 86M | 8M | 0.568 | 8.95 | 29.89 | 21.60 |
| | TEMPO (Zero Shot) | 345M | 8M | 0.598 | 10.54 | 26.69 | 22.30 |
| | TimesFM (Zero Shot) | 200M | 5M | 0.574 | 7.38 | 28.50 | 19.56 |
| **Supervised Models** | DLinear | 700K | 0 | 0.567 | 8.13 | 29.68 | 23.18 |
| | Auto-ARIMA | 10 | 0 | 0.896 | 12.82 | 0.00 | 0.00 |
| | AR | 513 | 0 | 0.556 | 6.57 | 31.17 | 24.31 |
| | **Auto-AR (our workflow)** | 513 | 0 | 0.551 | 5.45 | 31.91 | **25.36** |

Table 3: Aggregate forecasting performance across seven tasks, demonstrating that simply tuning a classical AR model is competitive with state-of-the-art FMs while using no pretraining data and tens of thousands of times fewer parameters. For Mean / Median %Imp. we report % improvement over Auto-ARIMA, and for Auto-AR, the model size refers to the largest configuration across tasks.

than pretraining and because its multi-task results *underperform* its single-task method, which if anything supports our claim about the limited (realized) benefits of transfer learning in time series.

We focus on long-horizon forecasting, which has a standard set of datasets (Goswami et al., 2024, Table 11), of which we consider seven.[5] Each consists of four settings corresponding to different time horizons, so in total this yields twenty-eight tasks. We compute aggregate metrics using RMSE, not MSE, so that performance scales linearly with prediction error; this choice has no effect on average rank. Lastly, note that since our focus is on *supervised* baselines, our main comparison with zero-shot models (Table 16) is in a less challenging setting than the one they report numbers for. However, AR can also be used in the ZS setting by training on just the history provided for the test series; as reported in Section 5.5, this is competitive with open-source ZS FMs at short time horizons.

**Baselines:**  As baselines we mainly use linear forecasters, including the classical (untuned) linear auto-regression (AR), the automated workhorse Auto-ARIMA (Hyndman & Khandakar, 2008), the more recent DLinear (Zeng et al., 2023), and our own workflow Auto-AR described in Section 3.2. Lastly, we also evaluate our other approach, DASHA, on six of the tasks (c.f. Table 14).

**Results:**  Table 16 shows that on the full seven-dataset evaluation our Auto-AR workflow always attains competitive performances across all aggregate metrics considered, and in particular attains the best median improvement over Auto-ARIMA. Specifically, our Auto-AR workflow achieves competitive performances with two other recent time series FMs, namely MOMENT and S$^2$IP-LLM, and outperforms the rest of the FMs. Although TTM surpasses all other methods across three aggregated metrics, the improvements remain relatively marginal. Thus, in this the domain as well we find that increases in pretraining dataset size and model scale have not yet resulted in significant performance improvements. Notably, even *untuned* AR, which uses no differencing and a large lookback window, is quite effective, doing better than zero-and-few-shot FMs across all metrics.

## 5 DISCUSSION

At a high level, our results show that the foundation models in these three domains have not yet surpassed supervised learning, and thus more broadly that the latter remains a strong baseline for specialized FMs. This is a surprising and consequential finding due the paradigm's popularity and the data and compute costs associated with large-scale pretraining. In this section we discuss lessons and implications for the development of machine learning in these and other application areas.

### 5.1 THE IMPORTANCE OF DIVERSE, WELL-TUNED, AND DOMAIN-SPECIFIC BASELINES

The main lesson of our work is to select a diverse array of baselines, drawing from both "lift-and-shift" and domain-specific approaches, and then to carefully tune them. For example, in genomics the vanilla wide ResNet baseline does remarkably well, with the majority of FMs doing worse than even this "lift-and-shift" baseline on the typical task in the NT benchmark. While satellite FMs

---

[5]The two we do not consider, Exchange and ILI, are not evaluated on by most time series FMs.

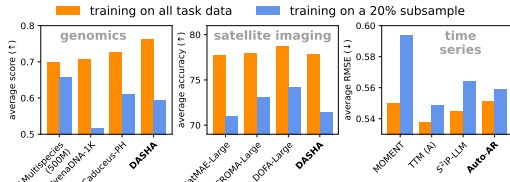

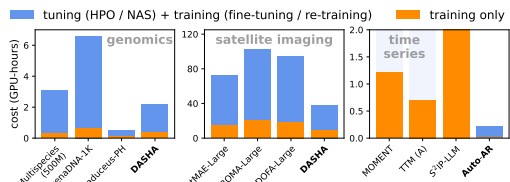

Figure 3: Performance of the best FMs when given only 20% as much fine-tuning data (c.f. Table 19). Supervised baselines are competitive even in this data-scarce regime: in satellite imaging and time series they are outperformed by just one FM, while in genomics they are beaten by two. Notably the worst of the three genomics FMs in the full setting does best with less data.

Figure 4: Tuning costs (excluding FM pre-training but including HPO for all methods and NAS for DASHA) and downstream fine-tuning / re-training costs (c.f. Table 18). Apart from Caduceus, supervised models are much cheaper, and for time series the entire Auto-AR pipeline is $3.5\times$ faster than fine-tuning any FM once (n.b. tuning costs for time series FMs are unknown).

do outperform such baselines, lightly modifying these CNNs via different kernel sizes and dilation rates was enough to match state-of-the-art models there as well. Lastly, our time series results demonstrate in dramatic fashion the need to carefully tune domain-specific approaches, as we show that simply allowing the classical AR forecaster to make use of long lookback windows and GPU-based optimization leads better forecasting than all open-source FMs.

## 5.2 THE IMPORTANCE OF BENCHMARKS THAT ADDRESS ACTUAL FM USE-CASES

A common argument in favor of foundation models is utility in low-data regimes, where transfer might be necessary to cope with limited supervision. However, to demonstrate usefulness via this argument, FMs must be evaluated in data-scarce regimes rather than the usual full-data settings they (and thus also we) generally consider. Furthermore, as demonstrated in Figure 3, our supervised baselines remain competitive even when only 20% of the data is provided to each task. Together with the zero-shot Auto-AR results (c.f. Section 5.5), this indicates that significant effort needs to be put in to designing benchmarks that elucidate the claimed potential of existing FMs.

## 5.3 COMPUTATIONAL EFFICIENCY CONSIDERATIONS

While computational efficiency is not our main focus, we nevertheless note that any performance gains from FMs must also be balanced against any extra cost. In addition to the usually massive costs of pretraining, the resulting models are often very big and costly to use. Indeed, apart from the special case of HyenaDNA and Caduceus, the CNN architectures discovered and trained using DASHA are typically over ten times smaller than FMs in the case of genomics and three to ten times smaller in the case of satellite imaging. Moreover, for time series our Auto-AR approach is quick-to-train and yields simple models with less than 1K parameters—over $2000\times$ smaller than any FM—while attaining performance that is often competitive even with closed-source models. We visualize the computational advantages of smaller supervised models in Figure 4, where we compare them against the top FMs in each domain. For genomics, fine-tuning most FMs takes roughly the same time as training our baselines, while for satellite imaging, FMs take roughly twice as long. Lastly, in time series our Auto-AR pipeline is over $30\times$ faster than large FMs and over three times faster than smaller models like TTM. This demonstrates efficiency of the supervised approaches sets a high performance bar that FMs need to clear before they can be deemed useful.

## 5.4 THE POWER OF TUNING KERNEL SIZES AND DILATION RATES

Our results for genomics and satellite imaging are driven by the DASHA workflow, whose crucial component is the tuning of kernel sizes and dilation rate in CNN backbones such as wide ResNet. Its success demonstrates that the procedure is an effective surrogate for human-driven model development, enabling the automated discovery of the types of diverse, domain-specific baselines stressed in Section 5.1. To understand this further, we study whether the architecture search component selects different kernel sizes and dilation rates for different tasks, and whether it does so in a consistent manner. Specifically, we run DASHA on three of the smaller datasets in the NT benchmark with fifteen different random seeds, construct eighteen-dimensional vectors of the discovered kernel sizes and dilation rates assigned to each of the nine layers, and project these to two dimensions using principal component analysis (PCA). The result in Figure 5 reveals that the architectures are clustered by task, demonstrating that the procedure selects different but consistent-within-task

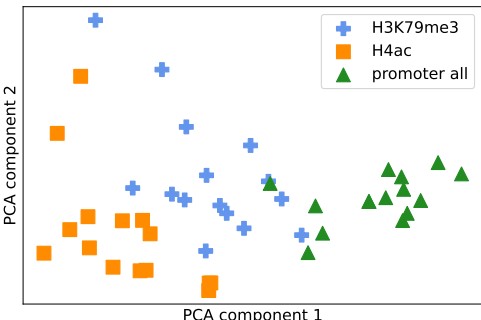

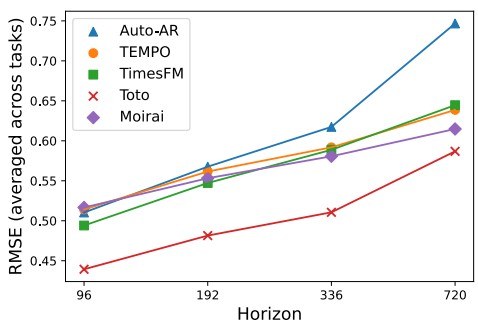

Figure 5: PCA visualization of architectures discovered for three different tasks when DASHA is run multiples times. Clustering across tasks reveals the within-task consistency of the architecture search component and the utility of diverse models as baselines.

Figure 6: Zero-shot performance on the subset of time series tasks with Moirai and Toto evaluations. At shorter horizons, Auto-AR trained on only the history given by the test series (and thus also given to ZS FMs) is competitive with all FMs except the closed-source Toto.

kernel parameters. This visualization suggests that architecture search is a useful surrogate for model development, and consequently that the DASHA workflow may also be useful for automating similar studies and baselining FMs in other domains with high-dimensional, unstructured data.

### 5.5 THE SURPRISING EFFECTIVENESS OF LINEAR AUTO-REGRESSION

Perhaps our most surprising finding is the competitiveness of linear auto-regression (AR), a very old method, on long-horizon forecasting. It is likely that the lack of comparison with this baseline was driven by existing evaluations (e.g. by Challu et al. (2022)) of Auto-ARIMA (Hyndman & Khandakar, 2008), which is *perceived* to be a stronger baseline because it both combines AR with another model (MA) and tunes the lookback and differencing parameters. However, in most Auto-ARIMA packages the default maximum lookback is around five, whereas we often found much (hundred-fold) larger settings to work best. Since these implementations are also generally too slow to support such long lookbacks, the possibility of expanding the hyperparameter space was more likely to be ignored. By implementing an efficient tuning procedure over a larger space of lookback parameters, our Auto-AR workflow comprises a significant contribution to forecasting baselines.

As described in Appendix F, Auto-AR also turns out to be easily extended to and competitive in the zero-shot setting, enabling a direct comparison with ZS FMs (c.f. Figure 6). The method involves training a short-lookback AR model on subsequences of the history of the length 512 provided in the ZS regime. This establishes what we believe should be the default supervised baseline for time series forecasting in the ZS setting. We provide full evaluations in Tables 20 & 21.

### 5.6 LIMITATIONS

While our findings are significant according to measures set by past work, they should not be misinterpreted to address all possible scenarios where FMs may be useful; for example, genomics FMs are often used for exploratory science and not prediction. We are also of course computationally limited and there are many other domains where FMs have been pretrained, and even in our three there are other tasks beyond classification and forecasting. Nevertheless, our evaluation is extensive—over fifty tasks and thirty FMs—and so the results provide at least a strongly indicator w.r.t. the state of a field that uses benchmark performance to motivate and justify pretraining.

## 6 CONCLUSION

We conduct a thorough investigation to evaluate whether the cost of training specialized foundation models across three major domains are justified by their superior performance relative to traditional supervised learning. Our results demonstrate that FMs in these domains have not yet surpassed supervised workflows and are often outperformed by fairly simple methods, including lightly modified CNN backbones (in genomics and satellite imaging) and classical linear forecasters (for time series). As part of our study, we introduce two automated workflows—**DASHA** for simulating in-domain model development of CNNs and **Auto-AR** for tuning linear auto-regression on GPUs—that we believe will be useful tools for evaluating future work in these and other areas. The code for these pipelines and to reproduce our results is publicly available.

## ACKNOWLEDGMENTS

We thank Mononito Goswami, Esther Rolf, and Stephan Xie for useful feedback. This work was supported in part by NSF grants IIS1705121, IIS1838017, IIS2046613, IIS2112471, a TCS Presidential Fellowship, and funding from Meta, Morgan Stanley, Amazon, Google, and Scribe. Any opinions, findings, and conclusions or recommendations expressed in this material are those of the author(s) and do not necessarily reflect the views of any of these funding agencies.

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

# A  TASKS

## A.1  GENOMICS

For the Genomics domain, we use the eighteen classification tasks from the Nucleotide Transformer benchmark (Dalla-Torre et al., 2023) that has widely been used for other genomics FMs. The benchmark datasets consist of nucleotide base sequences ranging from 200 to 600 bases in length. It provides a realistic and biological meaningful benchmark across four main categories: promoter (human/mouse), enhancer (human), splice site (SS; human/multispecies) and histone modification (yeast). Within the benchmark, the `enhancers_types` and `splice_sites_all` tasks are classification tasks with three classes each, while the remaining tasks are binary classification tasks.

| Dataset | # of classes | # of samples | | Maximum sequence length | Metric |
|---|---|---|---|---|---|
| | | train | test | | |
| enhancers | 2 | 14968 | 400 | 200 | MCC |
| enhancers_types | 3 | 14968 | 400 | 200 | MCC |
| promoter_all | 2 | 53276 | 5920 | 300 | F1 |
| promoter_no_tata | 2 | 47767 | 5299 | 300 | F1 |
| promoter_tata | 2 | 5509 | 621 | 300 | F1 |
| splice_sites_acceptors | 2 | 19961 | 2218 | 600 | F1 |
| splice_sites_all | 3 | 27000 | 3000 | 400 | Accuracy |
| splice_sites_donors | 2 | 19775 | 2198 | 600 | F1 |
| H3 | 2 | 13468 | 1497 | 500 | MCC |
| H3K14ac | 2 | 29743 | 3305 | 500 | MCC |
| H3K36me3 | 2 | 31392 | 3488 | 500 | MCC |
| H3K4me1 | 2 | 28509 | 3168 | 500 | MCC |
| H3K4me2 | 2 | 27614 | 3069 | 500 | MCC |
| H3K4me3 | 2 | 25953 | 2884 | 500 | MCC |
| H3K79me3 | 2 | 25953 | 2884 | 500 | MCC |
| H3K9ac | 2 | 25003 | 2779 | 500 | MCC |
| H4 | 2 | 13140 | 1461 | 500 | MCC |
| H4ac | 2 | 30685 | 3410 | 500 | MCC |

Table 4: Statistics for Genomics datasets

## A.2 SATELLITE IMAGING

In the satellite imaging domain, we aim to conduct evaluations with real-world relevance to Earth science. To achieve this, we include a variety of data from different sources to cover a diverse range of tasks, such as brick kiln identification, deforestation prediction, and photovoltaic monitoring. We utilize five classification tasks provided by the GeoBench dataset (Lacoste et al., 2024), a recently developed benchmark that offers a clean and carefully curated collection of tasks specifically designed for satellite imaging. In addition to GeoBench, we evaluate our model on three additional datasets (Helber et al., 2019; Jacques et al., 2023; Sumbul et al., 2019) commonly used in the literature as benchmarks for this domain . This brings the total to eight datasets, encompassing a wide range of features. These tasks vary in complexity, with single-class classification ranging from binary to 62-class problems, as well as two multilabel classification tasks. The datasets are further characterized by diverse input channels, ranging from 3 RGB channels to 18 channels that integrate data from both Sentinel-1 and Sentinel-2 formats.

For Geo-Bench datasets, we do not use any `mixup` and `cutmix` augmentations. For other datasets, we universally use `mixup = 0.8`, `cutmix = 1.0`, and a switch probability of 0.5. Following Fuller et al. (2023), we use only 10% of training set from BigEarthNet and fMoW-Sentinel while using the full evaluation set for validation.

| Dataset | Image Size | # of classes | # of samples | | | # of channels |
|---|---|---|---|---|---|---|
| | | | train | val | test | |
| m-bigearthnet | $120 \times 120$ | 43 | 20000 | 1000 | 1000 | 12 |
| m-brickkiln | $64 \times 64$ | 2 | 15063 | 999 | 999 | 13 |
| m-so2sat | $32 \times 32$ | 17 | 19992 | 986 | 986 | 18 |
| m-forestnet | $332 \times 332$ | 12 | 6464 | 989 | 993 | 6 |
| m-pv4ger | $320 \times 320$ | 2 | 11814 | 999 | 999 | 3 |
| BigEarthNet | $120 \times 120$ | 19 | 31166 | 103944 | 103728 | 12 |
| EuroSAT | $64 \times 64$ | 13 | 16200 | 10800 | 5400 | 13 |
| Canadian Cropland | $120 \times 120$ | 10 | 53884 | 11414 | 11674 | 12 |
| fMoW-Sentinel | $96 \times 96$ | 62 | 71287 | 84939 | 84966 | 13 |

Table 5: Statistics for Satellite datasets

## A.3 TIME SERIES

In the time series domain, we focus on the long horizon forecasting task. We use a subset of the common benchmark datasets for evaluating models across different domains (ETT, Electricity, Weather, Illness, Traffic, Exchange Rate) (Wang et al., 2024), specifically, the ETT, Weather, Electricity, Illness (ILI), and Traffic datasets. Note that the ETT dataset is actually a collection of four series: ETTh1, ETTh2, ETTm1, and ETTm2; we follow the rest of the literature in treating each series as a separate dataset. Each dataset contains measurements of one or more channels at evenly spaced time steps.

| Dataset | # of channels | # of samples | | |
|---|---|---|---|---|
| | | train | val | test |
| ETTh1 | 7 | 8033 | 2785 | 2785 |
| ETTh2 | 7 | 8033 | 2785 | 2785 |
| ETTm1 | 7 | 33953 | 11425 | 11425 |
| ETTm2 | 7 | 33953 | 11425 | 11425 |
| Weather | 21 | 36280 | 5175 | 10444 |
| Electricity | 321 | 17805 | 2537 | 5165 |
| ILI | 7 | 69 | 2 | 98 |
| Traffic | 862 | 11673 | 1661 | 3413 |

Table 6: Statistics for Time Series datasets

## B  IMPLEMENTATION DETAILS

### B.1  DASHA

Following the architecture search, we perform hyperparameter tuning using ASHA. The hyperparameter search space includes learning rate, weight decay, momentum, drop rate, and random seed for model initialization. We define a continuous search space, with further specific details provided in Table 7. Using ASHA, we evaluate 200 sample configurations over a maximum of 20 epochs, using a reduction factor of 2. The low-performing configurations are pruned based on their validation scores.

Before retraining the final model, we load the model checkpoint corresponding to the optimal hyperparameter configuration. The model is then trained for 200 epochs on the training data, with the best-performing checkpoint selected based on validation performance. This process is repeated for each backbone architecture, and the best-performing backbone is selected using the validation score. Finally, the checkpoint for the selected backbone is evaluated on the test set to obtain the final score.

| Hyperparameter | Search Space | Type of Search Space |
|---|---|---|
| random_seed | $[0, 500]$ | Integer |
| lr | $[10^{-5}, 5 \times 10^{-1}]$ | Log Uniform |
| drop_rate | $\{0, 0.05, 0.1\}$ | Discrete |
| weight_decay | $[5 \times 10^{-7}, 5 \times 10^{-3}]$ | Log Uniform |
| momentum | $[0.9, 1]$ | Uniform |

Table 7: Hyperparameter Search Space

### B.2  AUTO-AR

A fairly complete description is provided in Section 3.2. Here we note only that, because we minimize the total maximum likelihood across (independent) channels, to determine the amount of differencing used for each task we run the KPSS test separately on each channel and use the differencing needed by the majority of the channels. Notably, this results in a differencing of one for each task.

## C  DETAILED RESULTS

### C.1  GENOMICS

We include all FMs listed in Dalla-Torre et al. (2023, Supplementary Table 6) with addition of the two recently released models, Caduceus (Schiff et al., 2024) and Gena-LM (Fishman et al., 2024); note that the last row of tasks in the NT paper(promoter and splice sites) are mislabeled, but we infer an order in combination with previous information obtained from a (since-deleted) Huggingface leaderboard.[6] In alignment with the leaderboard, we apply a 0.1 validation split for DASHA during our evaluation. Additionally, we use an architecture set that includes both Wide ResNet and UNet for the search with DASHA on these datasets. We use batch size= 128 for all datasets, and cross entropy loss for all the training and finetuning. Individual scores for each task in the benchmark are provided in Tables 9 and 8.

The performance of the models on individual tasks is detailed in Tables 8 and 9. In the regulatory elements domain (c.f. Table 8), DASHA performs slightly behind the largest models like NT-Multispecies (Dalla-Torre et al., 2023) on the enhancers tasks but consistently outperforms models such as DNABERT-1 (Ji et al., 2021), Enformer (Avsec et al., 2021), and HyenaDNA (Nguyen et al., 2024); in the promoters task in generally performs worse than all reported FMs. In the RNA production domain, DASHA performs near the middle of the FMs. However, where DASHA truly excels is in the histone modification tasks (in Table 9), where it not only competes with, but often outperforms, the other FMs, consistently achieving top scores in nearly all tasks.

---

[6]See https://huggingface.co/spaces/InstaDeepAI/nucleotide_transformer_benchmark.

| Model | Regulatory Elements | | | | | RNA Production | | |
|---|---|---|---|---|---|---|---|---|
| | enhancers | enhancers types | promoter all | promoter no_tata | promoter tata | splice_sites acceptors | splice_sites all | splice_sites donors |
| Enformer | 0.454 | 0.312 | 0.955 | 0.955 | 0.959 | 0.915 | 0.847 | 0.906 |
| NT-1000G (500M) | 0.509 | 0.395 | 0.951 | 0.951 | 0.936 | 0.965 | 0.968 | 0.971 |
| NT-1000G (2.5B) | 0.546 | 0.432 | 0.965 | 0.967 | 0.957 | 0.98 | 0.976 | 0.979 |
| NT-Multispecies (500M) | 0.559 | 0.438 | 0.976 | 0.976 | 0.965 | 0.981 | 0.984 | 0.987 |
| NT-Multispecies (2.5B) | 0.545 | 0.444 | 0.975 | 0.977 | 0.959 | 0.986 | 0.982 | 0.987 |
| DNABERT-1 | 0.495 | 0.367 | 0.961 | 0.962 | 0.956 | – | 0.975 | – |
| DNABERT-2 | 0.525 | 0.423 | 0.972 | 0.972 | 0.955 | 0.975 | 0.939 | 0.963 |
| HyenaDNA-1K | 0.52 | 0.403 | 0.959 | 0.959 | 0.944 | 0.959 | 0.956 | 0.947 |
| HyenaDNA-32K | 0.489 | 0.352 | 0.956 | 0.954 | 0.939 | 0.96 | 0.962 | 0.957 |
| Caduceus-PS | 0.491 | 0.416 | 0.967 | 0.968 | 0.957 | 0.936 | 0.927 | 0.874 |
| Caduceus-PH | 0.546 | 0.439 | 0.97 | 0.969 | 0.953 | 0.937 | 0.94 | 0.948 |
| Wide ResNet | 0.525 | 0.416 | 0.952 | 0.946 | 0.93 | 0.821 | 0.457 | 0.815 |
| UNet | 0.49 | 0.366 | 0.956 | 0.954 | 0.95 | 0.956 | 0.955 | 0.968 |
| **DASHA** | 0.527 | 0.432 | 0.958 | 0.962 | 0.957 | 0.978 | 0.979 | 0.978 |

Table 8: Regulatory Elements and RNA Production tasks. "-" indicates unknown quantities.

| Model | H3 | H3K14ac | H3K36me3 | H3K4me1 | H3K4me2 | H3K4me3 | H3K79me3 | H3K9ac | H4 | H4ac |
|---|---|---|---|---|---|---|---|---|---|---|
| Enformer | 0.724 | 0.284 | 0.345 | 0.291 | 0.207 | 0.156 | 0.498 | 0.415 | 0.735 | 0.275 |
| NT-1000G (500M) | 0.736 | 0.381 | 0.468 | 0.38 | 0.26 | 0.235 | 0.562 | 0.479 | 0.755 | 0.342 |
| NT-1000G (2.5B) | 0.754 | 0.453 | 0.53 | 0.418 | 0.278 | 0.311 | 0.574 | 0.491 | 0.787 | 0.408 |
| NT-Multispecies (500M) | 0.786 | 0.549 | 0.624 | 0.55 | 0.32 | 0.406 | 0.63 | 0.567 | 0.799 | 0.496 |
| NT-Multispecies (2.5B) | 0.793 | 0.538 | 0.618 | 0.541 | 0.324 | 0.408 | 0.623 | 0.547 | 0.808 | 0.492 |
| DNABERT-1 | 0.763 | 0.403 | 0.474 | 0.396 | 0.282 | 0.258 | 0.578 | 0.505 | 0.784 | 0.359 |
| DNABERT-2 | 0.785 | 0.515 | 0.591 | 0.512 | 0.333 | 0.353 | 0.615 | 0.545 | 0.797 | 0.465 |
| HyenaDNA-1K | 0.781 | 0.608 | 0.614 | 0.512 | 0.455 | 0.55 | 0.669 | 0.586 | 0.763 | 0.564 |
| HyenaDNA-32K | 0.747 | 0.405 | 0.479 | 0.387 | 0.276 | 0.291 | 0.567 | 0.472 | 0.761 | 0.379 |
| Caduceus-PS | 0.799 | 0.541 | 0.609 | 0.488 | 0.388 | 0.44 | 0.679 | 0.604 | 0.789 | 0.525 |
| Caduceus-PH | 0.815 | 0.631 | 0.601 | 0.523 | 0.487 | 0.544 | 0.697 | 0.622 | 0.811 | 0.621 |
| Wide ResNet | 0.798 | 0.667 | 0.670 | 0.554 | 0.541 | 0.660 | 0.706 | 0.620 | 0.754 | 0.657 |
| UNet | 0.797 | 0.647 | 0.482 | 0.541 | 0.553 | 0.292 | 0.562 | 0.624 | 0.760 | 0.389 |
| **DASHA** | 0.790 | 0.683 | 0.630 | 0.528 | 0.640 | 0.714 | 0.721 | 0.709 | 0.776 | 0.744 |

Table 9: Histone Modification tasks

Note that because GENA-LM reports using MCC on all datasets, which is different than the metrics used by NT and Caduceus paper, we compare its results with DASHA in a separate table, where all datasets are evaluated through MCC. As shown in Table 10 and Table 11, DASHA also outperform GENA-LM by a large margin in terms of average MCC.

| Model | H3 | H3K14ac | H3K36me3 | H3K4me1 | H3K4me2 | H3K4me3 | H3K79me3 | H3K9ac | H4 | H4ac |
|---|---|---|---|---|---|---|---|---|---|---|
| GENA-LM | 0.79 | 0.6 | 0.61 | 0.53 | 0.46 | 0.55 | 0.67 | 0.61 | 0.78 | 0.59 |
| **DASHA** | 0.79 | 0.68 | 0.63 | 0.53 | 0.64 | 0.71 | 0.72 | 0.71 | 0.78 | 0.74 |

Table 10: Histone Modification for GENA-LM (all scores are obtained using MCC)

| Model | enhancers | enhancers types | promoter all | promoter no_tata | promoter tata | splice_sites acceptors | splice_sites all | splice_sites donors | Average |
|---|---|---|---|---|---|---|---|---|---|
| GENA-LM | 0.55 | 0.45 | 0.94 | 0.94 | 0.91 | 0.92 | 0.91 | 0.91 | 0.707 |
| **DASHA** | 0.53 | 0.43 | 0.92 | 0.92 | 0.90 | 0.97 | 0.96 | 0.96 | 0.755 |

Table 11: Regulatory Elements and RNA Production for GENA-LM. (all scores are obtained using MCC). The last column shows the average MCC across all 18 datasets in Tables 10 and 11.

## C.2 Satellite imaging

Training on satellite datasets requires relatively large computational resources due to the high number of channels and the size of the datasets. To ensure a fair comparison, we fine-tuned all the foundation models ourselves by sweeping across a fixed set of base learning rates $[5e-3, 2e-3, 2e-4, 4e-5]$. We then calculate the actual learning rate from base learning rate following previous work by $lr = base\_lr \cdot \frac{batch\_size}{256}$. This approach ensures that approximately the same amount of resources were used as during the DASHA tuning process, allowing for a balanced evaluation of model performance. For training and finetuning, we universally use `batch_size = 16` and loss function as cross entropy with 0.1 label smoothing for single label classification, and multi-label soft margin loss for multilabel classification. Individual scores for each task are provided in Table 12.

We closely followed the reported evaluation processes from previous studies on FMs (Cong et al., 2022; Fuller et al., 2023; Mendieta et al., 2023). These models do not employ a validation set for hyperparameter tuning or model selection, and we adhered to this same approach when fine-tuning the FMs. However, for DASHA, since we performed extensive hyperparameter optimization over a large search space, we used a validation set to ensure fair and accurate comparisons between DASHA and the FMs. This is a less favorable setting for DASHA, as it relies on extensive hyperparameter tuning, but we demonstrate that, even under these conditions, DASHA matches the performance of the FMs.

| Model | Average | m-bigearthnet | m-brickkiln | m-so2sat | m-forestnet | m-pv4ger | BigEarth Net | EuroSAT | Canadian Cropland | fMoW Sentinel |
|---|---|---|---|---|---|---|---|---|---|---|
| SatMAE-Base | 76.99 | 72.3 | 98.22 | 54.56 | 51.89 | 97.0 | 86.04 | 98.69 | 74.64 | 59.55 |
| SatMAE-Large | 77.75 | 73.82 | 98.6 | 55.79 | 53.7 | 96.92 | 86.75 | 98.86 | 75.38 | 59.89 |
| GFM | 77.18 | 71.97 | 98.35 | 57.52 | 59.38 | 96.54 | 85.93 | 99.02 | 72.03 | 53.84 |
| SwinT-Base | 76.69 | 70.14 | 98.81 | 56.49 | 59.78 | 97.54 | 85.91 | 98.99 | 70.15 | 52.37 |
| CROMA-Base | 77.39 | 72.07 | 98.99 | 60.04 | 54.07 | 96.6 | 86.94 | 98.81 | 75.87 | 53.14 |
| CROMA-Large | 78.03 | 73.36 | 99.01 | 59.22 | 51.96 | 96.9 | 87.98 | 98.98 | 76.56 | 58.32 |
| ResNet50 | 73.76 | 60.31 | 98.4 | 51.83 | 54.29 | 96.79 | 79.16 | 98.23 | 72 | 52.84 |
| Wide ResNet | 73.97 | 69.15 | 98.95 | 49.04 | 52.14 | 96.34 | 80.48 | 98.6 | 72.05 | 49.00 |
| UNet | 75.73 | 69.89 | 98.6 | 56.87 | 57.18 | 97.39 | 83.9 | 98.99 | 72.68 | 46.11 |
| **DASHA** | 77.85 | 72.72 | 98.92 | 56.28 | 57.24 | 97.4 | 86.09 | 99.07 | 75.69 | 57.20 |

Table 12: Satellite imaging tasks

It is also important to note that SatMAE only accepts 3-channel and 12-channel inputs, while CROMA is limited to 12-channel inputs. GeoBench, however, includes a wide range of tasks with varying numbers of input channels, ranging from 3 to 18. Despite these differences, we include all datasets in our evaluation because they are valuable benchmarks in the satellite image domain, and it is crucial for FMs in this field to generalize across diverse datasets. For datasets where the input size does not match the model requirements, we pad missing channels with zeros and prune any extra channels. However, to ensure a fair comparison, in addition to reporting the average scores across all datasets, we also provide average scores excluding m-pv4ger and m-forestnet, where missing channels may affect the performance of the FMs. The aggregate scores excluding m-pv4ger and m-forestnet are presented in Table 13.

| | Model | Average Score ↑ | Average Rank ↓ | Mean %Imp.↑ | Median %Imp.↑ |
|---|---|---|---|---|---|
| **Foundation Models** | SatMAE-Base | 77.71 | 6.00 | 6.74 | 4.56 |
| | SatMAE-Large | 78.44 | 4.07 | 7.87 | **6.75** |
| | GFM | 76.95 | 5.57 | 5.40 | 4.08 |
| | SwinT-Base | 76.12 | 6.50 | 3.98 | 1.43 |
| | CROMA-Base | 77.98 | 3.57 | 6.95 | 5.30 |
| | CROMA-Large | **79.06** | **2.14** | **8.84** | 6.26 |
| **Supervised Models** | ResNet50 | 73.25 | 9.00 | -0.27 | -0.38 |
| | Wide ResNet | 73.90 | 8.00 | 0.00 | 0.00 |
| | UNet | 75.29 | 6.57 | 2.33 | 0.87 |
| | **DASHA (our workflow)** | 78.00 | 3.57 | 7.02 | 5.16 |

Table 13: Aggregated metrics on satellite imaging tasks excluding the m-pv4ger and m-forestnet.

## C.3 Time series

Long horizon forecasting for time series is the following task: at every timestep $t$, take the $L$ historical observations at times $(t - L + 1, ..., t)$ for each channel and predict the next $H$ observations $(t+1, ..., t+H)$ for every channel. Following the literature, we evaluate on $H \in \{24, 36, 48, 60\}$ on ILI and $\{96, 192, 336, 720\}$ otherwise. For most methods, we report results for $L = 512$. All results are reported on a 70/10/20 train/validation/test split for each datasets, except for the ETT datasets which have predefined splits. MSE is reported after all datasets have been scaled by the mean and variance of the training data.

In addition to the results for DASHA, Auto-AR, DLinear, and the FMs, we evaluate the performance of two simple baselines:

1. **AR (d=0):** The vanilla autoregressive model (Box & Jenkins, 1976) predicts the (scalar) value of a time series at $t + 1$ as a linear combination of the last $L$ timesteps and a constant, i.e. $\hat{x}_{t+1} = \alpha_0 + \alpha_1 x_t + \alpha_2 x_{t-1} + ... + \alpha_L x_{t-L+1}$ for learnable parameters $\alpha_0, ..., \alpha_L$. We fit these parameters using standard maximum likelihood techniques. Here "d=0" denotes no differencing.

2. **Auto-ARIMA:** ARIMA is a statistical method used for time series forecasting that combines three components: AutoRegressive (AR), Integrated (I), and Moving Average (MA). The AR component models the relationship between an observation and its lagged (past) values, assuming that past values have a linear influence on future ones. The Integrated component applies differencing to the data to remove trends or seasonality, making the time series stationary by stabilizing its mean over time. The MA component models the relationship between an observation and the residual errors from a moving average model applied to previous observations. ARIMA is characterized by three parameters: p (the number of lag observations), d (the number of differencing steps to achieve stationarity), and q (the number of lagged forecast errors). This model is particularly effective for univariate time series forecasting where patterns like trends or seasonality are present. To tune these parameters we use the popular Auto-ARIMA approach of Hyndman & Khandakar (2008) discussed in Section 3.2.

As described, all of our baselines can handle only univariate time series, while all of the benchmark datasets are multivariate (multiple channels). These baselines are trained under channel independence: each channel of a time series is treated independently. While channel independence fails to take into account cross-channel dependencies, we note that developing methods that leverage cross-channel dependencies for a variable number of channels remains an open problem.

### C.3.1 Time-LLM

Due to differences in the results reported for Time-LLM between the original paper and reproductions in several works (Goswami et al., 2024; Ekambaram et al., 2024; Pan et al., 2024; Tan et al., 2024), we to attempt our own partial reproduction in order to determine whether to include their numbers. To do so, we used code provided by the authors and looked at ETTh1 with input lengths from $96, 192$ and ILI with input lengths from $24, 36, 48, 90$ (going beyond this was infeasible due to the resources required to finetune LLaMA-7B). Our reproduced MSE for ETTh1 with an input length of 96 was $0.405$ as compared to that of $0.362$ reported by Jin et al. (2024), while for input length of 192 it was $0.431$ vs. $0.398$. On ILI the average increase in MSE across time horizons from their reported results to our reproduction was more than $0.4$. Due to these large discrepancies we chose to exclude their model from our analysis.

### C.3.2 Closed-source models

In Table 14 we report numbers for three additional FMs that either do not report a complete set of results (Moirai) on all seven datasets or are closed-source (LLM4TS) or both (Toto). We report aggregate metrics for a six dataset setting in Table 17, where we see that Toto dominates (despite being zero-shot) while the performance of Auto-AR is comparable to that of LLM4TS. A look at Figure 7 reveals that Toto is not superior across all six tasks, with the aggregated metrics being strongly boosted by its dramatically better performance on one of them (ETTh2). Thus, while its ZS performance is quite good, it is unclear whether this result would be maintained with additional tasks.

| Model | ETTh1 | | | | ETTh2 | | | | ETTm1 | | | | ETTm2 | | | |
|---|---|---|---|---|---|---|---|---|---|---|---|---|---|---|---|---|
| Horizon | 96 | 192 | 336 | 720 | 96 | 192 | 336 | 720 | 96 | 192 | 336 | 720 | 96 | 192 | 336 | 720 |
| GPT4TS (OFA) | 0.376 | 0.416 | 0.442 | 0.477 | 0.285 | 0.354 | 0.373 | 0.406 | 0.292 | 0.332 | 0.366 | 0.417 | 0.173 | 0.229 | 0.286 | 0.378 |
| TEST (Few shot) | 0.455 | 0.572 | 0.611 | 0.723 | 0.332 | 0.401 | 0.408 | 0.459 | 0.392 | 0.423 | 0.471 | 0.552 | 0.233 | 0.303 | 0.359 | 0.452 |
| LLM4TS | 0.371 | 0.403 | 0.42 | 0.422 | 0.269 | 0.328 | 0.353 | 0.383 | 0.285 | 0.324 | 0.353 | 0.408 | 0.165 | 0.22 | 0.268 | 0.35 |
| MOMENT | 0.387 | 0.41 | 0.422 | 0.454 | 0.288 | 0.349 | 0.369 | 0.403 | 0.293 | 0.326 | 0.352 | 0.405 | 0.17 | 0.227 | 0.275 | 0.363 |
| TTM (B) | 0.36 | 0.392 | 0.401 | 0.436 | 0.269 | 0.336 | 0.359 | 0.39 | 0.291 | 0.325 | 0.363 | 0.419 | 0.164 | 0.219 | 0.277 | 0.35 |
| TTM (A) | 0.363 | 0.392 | 0.413 | 0.442 | 0.262 | 0.324 | 0.351 | 0.392 | 0.283 | 0.332 | 0.353 | 0.393 | 0.158 | 0.213 | 0.269 | 0.369 |
| $S^2$IP-LLM | 0.366 | 0.401 | 0.412 | 0.44 | 0.278 | 0.346 | 0.367 | 0.4 | 0.288 | 0.323 | 0.359 | 0.403 | 0.165 | 0.222 | 0.277 | 0.363 |
| CALF | 0.369 | 0.427 | 0.456 | 0.479 | 0.279 | 0.353 | 0.362 | 0.404 | 0.323 | 0.374 | 0.409 | 0.477 | 0.178 | 0.242 | 0.307 | 0.397 |
| TEMPO (Zero Shot) | 0.4 | 0.426 | 0.441 | 0.443 | 0.301 | 0.355 | 0.379 | 0.409 | 0.438 | 0.461 | 0.515 | 0.591 | 0.185 | 0.243 | 0.309 | 0.386 |
| TimesFM (Zero Shot) | 0.421 | 0.472 | 0.51 | 0.514 | 0.326 | 0.399 | 0.434 | 0.451 | 0.357 | 0.411 | 0.441 | 0.507 | 0.205 | 0.294 | 0.367 | 0.473 |
| Moirai (Zero Shot) | 0.375 | 0.399 | 0.412 | 0.413 | 0.281 | 0.34 | 0.362 | 0.38 | 0.404 | 0.435 | 0.462 | 0.49 | 0.205 | 0.261 | 0.319 | 0.415 |
| Toto (Zero Shot) | 0.307 | 0.329 | 0.396 | 0.419 | 0.093 | 0.135 | 0.16 | 0.294 | 0.306 | 0.328 | 0.39 | 0.463 | 0.2 | 0.269 | 0.264 | 0.354 |
| DLinear | 0.375 | 0.405 | 0.439 | 0.472 | 0.289 | 0.383 | 0.448 | 0.605 | 0.299 | 0.335 | 0.369 | 0.425 | 0.167 | 0.224 | 0.281 | 0.397 |
| Auto-ARIMA | 0.646 | 0.704 | 0.732 | 0.738 | 0.324 | 0.411 | 0.456 | 0.462 | 1.131 | 1.172 | 1.197 | 1.231 | 0.225 | 0.298 | 0.37 | 0.478 |
| AR (d=0) | 0.358 | 0.39 | 0.41 | 0.424 | 0.271 | 0.334 | 0.361 | 0.395 | 0.299 | 0.336 | 0.368 | 0.426 | 0.163 | 0.218 | 0.271 | 0.366 |
| **Auto-AR** | 0.357 | 0.39 | 0.41 | 0.422 | 0.269 | 0.332 | 0.359 | 0.394 | 0.299 | 0.336 | 0.368 | 0.426 | 0.163 | 0.218 | 0.271 | 0.367 |
| **DASHA** | 0.369 | 0.401 | 0.430 | 0.478 | 0.284 | 0.377 | 0.396 | 0.745 | 0.305 | 0.335 | 0.367 | 0.418 | 0.169 | 0.224 | 0.290 | 0.378 |

| Model | Weather | | | | Electricity | | | | ILI | | | | Traffic | | | |
|---|---|---|---|---|---|---|---|---|---|---|---|---|---|---|---|---|
| Horizon | 96 | 192 | 336 | 720 | 96 | 192 | 336 | 720 | 96 | 192 | 336 | 720 | 96 | 192 | 336 | 720 |
| GPT4TS (OFA) | 0.162 | 0.204 | 0.254 | 0.326 | 0.139 | 0.153 | 0.169 | 0.206 | 2.063 | 1.868 | 1.79 | 1.979 | 0.388 | 0.407 | 0.412 | 0.45 |
| TEST (Few shot) | 0.163 | 0.23 | 0.278 | 0.301 | 0.143 | 0.158 | 0.176 | 0.23 | - | - | - | - | 0.415 | 0.425 | 0.436 | 0.489 |
| LLM4TS | 0.147 | 0.191 | 0.241 | 0.313 | 0.128 | 0.146 | 0.163 | 0.2 | – | – | – | – | 0.372 | 0.391 | 0.405 | 0.437 |
| MOMENT | 0.154 | 0.197 | 0.246 | 0.315 | 0.136 | 0.152 | 0.167 | 0.205 | 2.728 | 2.669 | 2.728 | 2.883 | 0.391 | 0.404 | 0.414 | 0.45 |
| TTM (B) | 0.146 | 0.19 | 0.242 | 0.323 | 0.129 | 0.149 | 0.163 | 0.2 | - | - | - | - | 0.368 | 0.403 | 0.395 | 0.431 |
| TTM (A) | 0.149 | 0.192 | 0.24 | 0.318 | 0.128 | 0.144 | 0.162 | 0.191 | - | - | - | - | 0.352 | 0.359 | 0.375 | 0.419 |
| $S^2$IP-LLM | 0.145 | 0.19 | 0.243 | 0.312 | 0.135 | 0.149 | 0.167 | 0.2 | - | - | - | - | 0.379 | 0.397 | 0.407 | 0.44 |
| CALF | 0.164 | 0.214 | 0.269 | 0.355 | 0.145 | 0.161 | 0.175 | 0.222 | - | - | - | - | 0.407 | 0.43 | 0.444 | 0.477 |
| TEMPO (Zero Shot) | 0.211 | 0.254 | 0.292 | 0.37 | 0.178 | 0.198 | 0.209 | 0.279 | - | - | - | - | 0.476 | 0.496 | 0.503 | 0.538 |
| TimesFM (Zero Shot) | 0.122 | 0.169 | 0.242 | 0.391 | 0.119 | 0.137 | 0.158 | 0.206 | - | - | - | - | 0.327 | 0.353 | 0.378 | 0.42 |
| Moirai (Zero Shot) | 0.173 | 0.216 | 0.26 | 0.32 | 0.205 | 0.22 | 0.236 | 0.27 | – | – | – | – | – | – | – | – |
| Toto (Zero Shot) | 0.18 | 0.235 | 0.252 | 0.356 | 0.124 | 0.138 | 0.155 | 0.211 | – | – | – | – | – | – | – | – |
| DLinear | 0.176 | 0.22 | 0.265 | 0.323 | 0.14 | 0.153 | 0.169 | 0.203 | 2.215 | 1.963 | 2.13 | 2.368 | 0.41 | 0.423 | 0.436 | 0.466 |
| Auto-ARIMA | 0.217 | 0.263 | 0.33 | 0.425 | 1.22 | 1.264 | 1.311 | 1.364 | 5.554 | 6.94 | 7.192 | 6.648 | 1.997 | 2.044 | 2.096 | 2.138 |
| AR (d=0) | 0.171 | 0.215 | 0.263 | 0.332 | 0.138 | 0.153 | 0.17 | 0.212 | 2.084 | 2.04 | 2.004 | 2.011 | 0.398 | 0.413 | 0.426 | 0.464 |
| **Auto-AR** | 0.172 | 0.215 | 0.263 | 0.332 | 0.138 | 0.153 | 0.17 | 0.212 | 2.084 | 2.04 | 2.004 | 2.011 | 0.398 | 0.413 | 0.426 | 0.464 |
| **DASHA** | 0.163 | 0.205 | 0.251 | 0.314 | 0.136 | 0.151 | 0.165 | 0.200 | – | – | – | – | – | – | – | – |

Table 14: Time series forecasting tasks (MSE). "-" indicates unknown quantities.

| Model | ETTh1 | | | | ETTh2 | | | | ETTm1 | | | | ETTm2 | | | |
|---|---|---|---|---|---|---|---|---|---|---|---|---|---|---|---|---|
| Horizon | 96 | 192 | 336 | 720 | 96 | 192 | 336 | 720 | 96 | 192 | 336 | 720 | 96 | 192 | 336 | 720 |
| GPT4TS (OFA) | 0.397 | 0.418 | 0.433 | 0.456 | 0.342 | 0.389 | 0.407 | 0.441 | 0.346 | 0.372 | 0.394 | 0.421 | 0.262 | 0.301 | 0.341 | 0.401 |
| TEST (Few shot) | 0.457 | 0.519 | 0.531 | 0.594 | 0.374 | 0.433 | 0.44 | 0.48 | 0.401 | 0.426 | 0.444 | 0.501 | 0.262 | 0.302 | 0.341 | 0.419 |
| LLM4TS | 0.394 | 0.412 | 0.422 | 0.444 | 0.332 | 0.377 | 0.396 | 0.425 | 0.343 | 0.366 | 0.385 | 0.419 | 0.254 | 0.292 | 0.326 | 0.38 |
| MOMENT | 0.41 | 0.426 | 0.437 | 0.472 | 0.345 | 0.386 | 0.408 | 0.439 | 0.349 | 0.368 | 0.384 | 0.416 | 0.26 | 0.297 | 0.328 | 0.387 |
| TTM (B) | - | - | - | - | - | - | - | - | - | - | - | - | - | - | - | - |
| TTM (A) | - | - | - | - | - | - | - | - | - | - | - | - | - | - | - | - |
| $S^2$IP-LLM | 0.396 | 0.42 | 0.431 | 0.458 | 0.34 | 0.385 | 0.406 | 0.436 | 0.346 | 0.365 | 0.39 | 0.418 | 0.257 | 0.299 | 0.33 | 0.39 |
| CALF | 0.389 | 0.423 | 0.436 | 0.467 | 0.331 | 0.38 | 0.394 | 0.426 | 0.349 | 0.375 | 0.399 | 0.438 | 0.256 | 0.297 | 0.339 | 0.393 |
| TEMPO (Zero Shot) | 0.406 | 0.421 | 0.43 | 0.451 | 0.353 | 0.389 | 0.408 | 0.44 | 0.424 | 0.432 | 0.467 | 0.509 | 0.267 | 0.304 | 0.345 | 0.395 |
| TimesFM (Zero Shot) | 0.421 | 0.472 | 0.51 | 0.514 | 0.326 | 0.399 | 0.434 | 0.451 | 0.357 | 0.411 | 0.441 | 0.507 | 0.205 | 0.294 | 0.367 | 0.473 |
| Moirai (Zero Shot) | 0.402 | 0.419 | 0.429 | 0.444 | 0.334 | 0.373 | 0.393 | 0.416 | 0.383 | 0.402 | 0.416 | 0.437 | 0.282 | 0.318 | 0.355 | 0.41 |
| Toto (Zero Shot) | 0.366 | 0.368 | 0.399 | 0.424 | 0.197 | 0.231 | 0.260 | 0.355 | 0.328 | 0.353 | 0.389 | 0.429 | 0.27 | 0.315 | 0.319 | 0.374 |
| DLinear | 0.399 | 0.416 | 0.443 | 0.49 | 0.353 | 0.418 | 0.465 | 0.551 | 0.343 | 0.365 | 0.386 | 0.421 | 0.26 | 0.303 | 0.342 | 0.421 |
| Auto-ARIMA | 0.537 | 0.566 | 0.587 | 0.608 | 0.368 | 0.418 | 0.454 | 0.467 | 0.659 | 0.683 | 0.701 | 0.725 | 0.296 | 0.34 | 0.381 | 0.438 |
| **Auto-AR** | 0.388 | 0.41 | 0.423 | 0.448 | 0.334 | 0.376 | 0.402 | 0.437 | 0.343 | 0.364 | 0.384 | 0.418 | 0.251 | 0.29 | 0.325 | 0.381 |

| Model | Weather | | | | Electricity | | | | ILI | | | | Traffic | | | |
|---|---|---|---|---|---|---|---|---|---|---|---|---|---|---|---|---|
| Horizon | 96 | 192 | 336 | 720 | 96 | 192 | 336 | 720 | 96 | 192 | 336 | 720 | 96 | 192 | 336 | 720 |
| GPT4TS (OFA) | 0.212 | 0.248 | 0.286 | 0.337 | 0.238 | 0.251 | 0.266 | 0.297 | 0.881 | 0.892 | 0.884 | 0.957 | 0.282 | 0.29 | 0.294 | 0.312 |
| LLM4TS | 0.196 | 0.238 | 0.277 | 0.329 | 0.223 | 0.24 | 0.258 | 0.292 | - | - | - | - | 0.259 | 0.265 | 0.275 | 0.292 |
| TEST (Few shot) | 0.213 | 0.263 | 0.282 | 0.328 | 0.235 | 0.255 | 0.275 | 0.311 | - | - | - | - | 0.317 | 0.3 | 0.31 | 0.338 |
| MOMENT | 0.209 | 0.248 | 0.285 | 0.336 | 0.233 | 0.247 | 0.264 | 0.295 | 1.114 | 1.092 | 1.098 | 1.126 | 0.282 | 0.287 | 0.292 | 0.31 |
| TTM (B) | - | - | - | - | - | - | - | - | - | - | - | - | - | - | - | - |
| TTM (A) | - | - | - | - | - | - | - | - | - | - | - | - | - | - | - | - |
| $S^2$IP-LLM | 0.195 | 0.235 | 0.28 | 0.326 | 0.23 | 0.247 | 0.266 | 0.287 | - | - | - | - | 0.274 | 0.282 | 0.289 | 0.301 |
| CALF | 0.204 | 0.25 | 0.291 | 0.352 | 0.238 | 0.252 | 0.267 | 0.303 | - | - | - | - | 0.268 | 0.278 | 0.281 | 0.3 |
| TEMPO (Zero Shot) | 0.254 | 0.298 | 0.332 | 0.379 | 0.276 | 0.293 | 0.309 | 0.355 | - | - | - | - | 0.343 | 0.355 | 0.356 | 0.376 |
| TimesFM (Zero Shot) | 0.122 | 0.169 | 0.242 | 0.391 | 0.119 | 0.137 | 0.158 | 0.206 | - | - | - | - | 0.327 | 0.353 | 0.378 | 0.42 |
| Moirai (Zero Shot) | 0.212 | 0.25 | 0.282 | 0.322 | 0.299 | 0.31 | 0.323 | 0.347 | - | - | - | - | - | - | - | - |
| Toto (Zero Shot) | 0.223 | 0.267 | 0.291 | 0.356 | 0.212 | 0.232 | 0.249 | 0.291 | - | - | - | - | - | - | - | - |
| DLinear | 0.237 | 0.282 | 0.319 | 0.362 | 0.237 | 0.249 | 0.267 | 0.301 | 1.081 | 0.963 | 1.024 | 1.096 | 0.282 | 0.287 | 0.296 | 0.315 |
| Auto-ARIMA | 0.267 | 0.341 | 0.426 | 0.53 | 0.441 | 0.468 | 0.485 | 0.54 | - | - | - | - | 0.753 | 0.819 | 0.932 | 1.243 |
| **Auto-AR** | 0.223 | 0.26 | 0.294 | 0.341 | 0.233 | 0.247 | 0.265 | 0.301 | 0.944 | 0.948 | 0.947 | 0.958 | 0.28 | 0.286 | 0.294 | 0.319 |

Table 15: Time series forecasting tasks (MAE). "-" indicates unknown quantities.

|  | Model | Average MAE ↓ | Average Rank ↓ | Mean %Imp.↑ | Median %Imp.↑ |
|---|---|---|---|---|---|
| **Foundation Models** | GPT4TS | 0.337 | 4.54 | 33.01 | 30.07 |
|  | TEST (Few Shot) | 0.370 | 6.36 | 27.35 | 26.89 |
|  | MOMENT | 0.336 | 3.68 | 33.32 | 30.19 |
|  | S²IP-LLM | 0.331 | 2.54 | 34.22 | 32.68 |
|  | CALF | 0.335 | 3.68 | 33.50 | 29.63 |
| **Supervised Models** | DLinear | 0.350 | 5.61 | 29.87 | 26.10 |
|  | Auto-ARIMA | 0.553 | 7.82 | 0.00 | 0.00 |
|  | **Auto-AR (our workflow)** | 0.333 | 2.71 | 33.86 | 29.46 |

Table 16: Aggregate performance on time series tasks across seven tasks based on MAE.

|  | Model Size | Pretraining Series | Average RMSE ↓ | Mean %Imp.↑ | Median %Imp.↑ |
|---|---|---|---|---|---|
| Moirai (Zero Shot) | 14M | 6M | 0.566 | 24.53 | 18.52 |
| LLM4TS | 60M | 8M | 0.526 | 29.25 | 20.96 |
| TTM (A) | 5M | 1M | 0.525 | 29.38 | 19.87 |
| Toto (Zero Shot) | 103M | - | **0.505** | **32.40** | **31.35** |
| **Auto-AR** | 513 | 0 | 0.534 | 28.12 | 19.63 |
| **DASHA** | 480K | 0 | 0.549 | 25.94 | 16.78 |

Table 17: Aggregate results for the excluded time series FMs in the six-dataset setting discussed in Appendix C.3.2. "-" indicates unknown quantities.

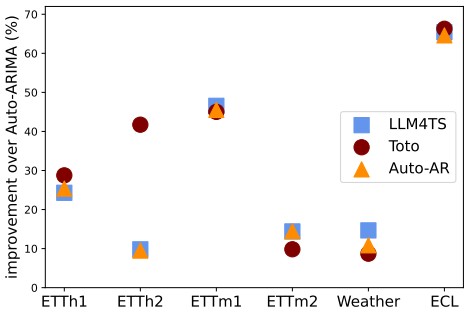

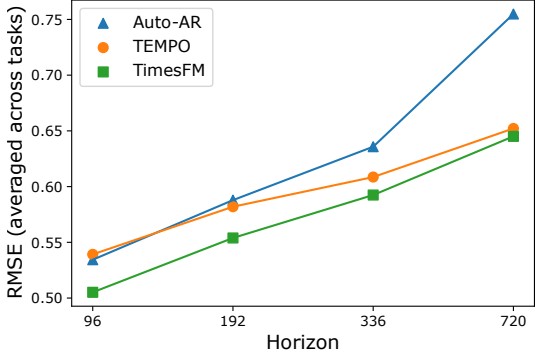

Figure 7: Scatterplot of improvement in RMSE (averaged across time horizons) of two closed-source FMs, LLM4TS and Toto, on a subset of datasets in Table 14. It shows that both FMs usually perform similarly to the Auto-AR baseline, with Toto attaining strong aggregate metrics due to a dominant performance on ETTh2.

Figure 8: Zero-shot performance comparisons on all time series datasets, as measured by the average RMSE across tasks. As in Figure 6, even in the ZS regime the Auto-AR baseline is competitive at the shorter time horizons with all open-source FMs.

## D  COMPUTATIONAL COSTS

Here we study the costs of our automated baselines against the top three FMs. The aggregated results for each domain are shown in Table 18, along with an explanation for how they were obtained. For genomics, fine-tuning FMs takes approximately the same time as training our baseline ( 0.36 GPU-hours). Developing and tuning our models takes 1.82 GPU-hours, which is better than all FMs except Caduceus-PH. For satellite data, both fine-tuning FMs and optimizing the hyperparameters of the fine-tuning procedure takes approximately $2\times$ longer than training our baselines. Lastly, for time series data our full pipeline is more than 30 times faster than larger language model (LLM)-based FMs such as $S^2$IP-LLM and more than 3 times faster than TTM.

| | model | GPU Hours | |
| | | model development + tuning | fine-tuning |
|---|---|---|---|
| **Genomics** | NT-Multispecies (500M) | 2.8 | 0.28 |
| | HyenaDNA-1K | 6.0 | 0.60 |
| | Caduceus-PH | 0.40 | 0.10 |
| | **DASHA** | 1.82 | 0.36 |
| **Satellite Imaging** | SatMAE-Large | 57.60 | 14.40 |
| | CROMA-Large | 81.88 | 20.47 |
| | DOFA-Large | 75.04 | 18.76 |
| | **DASHA** | 29.24 | 8.97 |
| **Time Series** | MOMENT | - | 1.22 |
| | TTM (A) | - | 0.71 |
| | $S^2$IP-LLM | - | $> 24$ |
| | **Auto-AR** | 0.20 | 0.01 |

Table 18: Computational costs of top-performing FMs and our methods (DASHA and AutoAR). All experiments were conducted on L40 GPUs (L40S for Genomics). For FMs, model development time only considers the time taken for hyperparameter tuning, and does not count the time taken to pre-train the models. "-" means that the model tuning time is unknown. Similarly, for NT-Multispecies (500M) and HyenaDNA-1K, the model tuning time is derived by 10 times the fine-tuning time, based on Section A.1.4 in Dalla-Torre et al. (2023). For satellite models, the model tuning time is similarly derived by reporting 4 times the fine-tuning time, which is the size of the hyperparameter sweep that we used. While we do not have good tuning estimates for time series, based on the above it is reasonable to assume it would take at least as long as fine-tuning the model once, which informs the pale blue bars in the right plot of Figure 4.

# E  SUBSAMPLING EXPERIMENTS

A common argument in favor of FMs is that they perform better in data-limited regime. Here, we examine the sensitivity of our results to downscaling the amount of available data. Specifically, for each domain-specific dataset, we use only 20% of the original data to train the supervised learning baseline and fine-tune the two best-performing foundation models in each domain. For satellite and genomics, the subset of the data is randomly selected from the original training dataset. For time series, the data are selected from the last 20% of the training series. We display the mean performance on all datasets in each domain in Table 19. The result demonstrate that modern FMs struggle to outperform simple supervised baselines even in data-limited regimes.

| | model | 100% | 20% | metric |
|---|---|---|---|---|
| **Genomics** | NT-Multispecies (500M) | 0.700 | 0.656 | average score (↑) |
| | HyenaDNA-1K | 0.708 | 0.516 | |
| | Caduceus-Ph | 0.725 | 0.610 | |
| | **DASHA** | 0.761 | 0.595 | |
| **Satellite Imaging** | SatMAE-Large | 77.75 | 70.96 | average accuracy (↑) |
| | CROMA-Large | 78.03 | 73.13 | |
| | DOFA-Large | 78.80 | 74.26 | |
| | **DASHA** | 77.85 | 71.49 | |
| **Time Series** | MOMENT | 0.550 | 0.594 | average RMSE (↓) |
| | TTM (A) | 0.538 | 0.549 | |
| | $S^2$IP-LLM | 0.545 | 0.564 | |
| | **Auto-AR** | 0.551 | 0.559 | |

Table 19: Subsampling experiments on 20% subset of finetuning data: the aggregated performances are obtained by averaging individual performances of models on different datasets in each domains. Overall, our supervised methods remain competitive even in the limited training data setting.

# F    ZERO-SHOT AUTO-AR

In this section, we describe how to run Auto-AR in a zero-shot manner, enabling a direct and fair comparison with ZS time series FMs. This approach is a check on the performance of such models, serving as a minimal baseline that they should be expected to overcome.

Given an input time series of length $L$ and prediction horizon $H$, the zero-shot Auto-AR methods works as follows:

1. segment the input into smaller fragments using a rolling window of size $W$ such that $W < L$, resulting in $L - W + 1$ samples of length $W$
2. run regular Auto-AR (the three steps described in Section 3.2) on these samples (for efficiency we restrict the search space over the lookback parameter to $\{64, 96, 128, 192\}$)
3. use the resulting model for prediction

Since this approach uses only the test example of length $L$ (e.g. $L = 512$), this approach uses the same data as is given to a ZS time series FM and so can be used as a fair baseline for the latter.

We evaluate ZS Auto-AR on all the tasks we consider and report the results in Tables 20 and 21. Aggregate performance is visualized in Figures 6 and 8, where we compare with the zero-shot forecasting performance of Auto-AR FMs such as TEMPO, TimesFM, Toto, and Moirai across different forecasting horizons. A key expectation is that ZS Auto-AR, trained on only a few chunks of temporal data, should struggle significantly compared to FMs, which are pretrained on millions of time series. However, at shorter horizons (96 and 192), Auto-AR performs competitively with foundation models, with only a marginal gap in average RMSE. Only at longer forecasting horizons does Auto-AR's performance deteriorates relative to that of FMs, with a widening performance gap. This suggests that ZS FMs do leverage pretraining to maintain robustness over longer horizons but are not significantly better than a simple baseline on short-horizon predictions (apart from Toto).

| Model | ETTh1 | | | | ETTh2 | | | | ETTm1 | | | | ETTm2 | | | |
|---|---|---|---|---|---|---|---|---|---|---|---|---|---|---|---|---|
| Horizon | 96 | 192 | 336 | 720 | 96 | 192 | 336 | 720 | 96 | 192 | 336 | 720 | 96 | 192 | 336 | 720 |
| TEMPO (Zero Shot) | 0.4 | 0.426 | 0.441 | 0.443 | 0.301 | 0.355 | 0.379 | 0.409 | 0.438 | 0.461 | 0.515 | 0.591 | 0.185 | 0.243 | 0.309 | 0.386 |
| TimesFM (Zero Shot) | 0.421 | 0.472 | 0.51 | 0.514 | 0.326 | 0.399 | 0.434 | 0.451 | 0.357 | 0.411 | 0.441 | 0.507 | 0.205 | 0.294 | 0.367 | 0.473 |
| Toto (Zero Shot) | 0.307 | 0.329 | 0.396 | 0.419 | 0.093 | 0.135 | 0.16 | 0.294 | 0.306 | 0.328 | 0.39 | 0.463 | 0.2 | 0.269 | 0.264 | 0.354 |
| Moirai (Zero Shot) | 0.375 | 0.399 | 0.412 | 0.413 | 0.281 | 0.34 | 0.362 | 0.38 | 0.404 | 0.435 | 0.462 | 0.49 | 0.205 | 0.261 | 0.319 | 0.415 |
| **Auto-AR (Zero Shot)** | 0.416 | 0.464 | 0.511 | 0.53 | 0.311 | 0.416 | 0.479 | 0.555 | 0.348 | 0.426 | 0.489 | 0.854 | 0.196 | 0.272 | 0.357 | 0.549 |

| Model | Weather | | | | Electricity | | | | ILI | | | | Traffic | | | |
|---|---|---|---|---|---|---|---|---|---|---|---|---|---|---|---|---|
| Horizon | 96 | 192 | 336 | 720 | 96 | 192 | 336 | 720 | 96 | 192 | 336 | 720 | 96 | 192 | 336 | 720 |
| TEMPO (Zero Shot) | 0.211 | 0.254 | 0.292 | 0.37 | 0.178 | 0.198 | 0.209 | 0.279 | - | - | - | - | 0.476 | 0.496 | 0.503 | 0.538 |
| TimesFM (Zero Shot) | 0.122 | 0.169 | 0.242 | 0.391 | 0.119 | 0.137 | 0.158 | 0.206 | - | - | - | - | 0.327 | 0.353 | 0.378 | 0.42 |
| Toto (Zero Shot) | 0.18 | 0.235 | 0.252 | 0.356 | 0.124 | 0.138 | 0.155 | 0.211 | - | - | - | - | - | - | - | - |
| Moirai (Zero Shot) | 0.173 | 0.216 | 0.26 | 0.32 | 0.205 | 0.22 | 0.236 | 0.27 | - | - | - | - | - | - | - | - |
| **Auto-AR (Zero Shot)** | 0.184 | 0.235 | 0.3 | 0.664 | 0.158 | 0.177 | 0.205 | 0.278 | 2.731 | 2.357 | 2.198 | 2.074 | 0.461 | 0.503 | 0.557 | 0.646 |

Table 20: Zero-shot results (MSE). "-" indicates unknown quantities.

| Model | ETTh1 | | | | ETTh2 | | | | ETTm1 | | | | ETTm2 | | | |
|---|---|---|---|---|---|---|---|---|---|---|---|---|---|---|---|---|
| Horizon | 96 | 192 | 336 | 720 | 96 | 192 | 336 | 720 | 96 | 192 | 336 | 720 | 96 | 192 | 336 | 720 |
| TEMPO (Zero Shot) | 0.406 | 0.421 | 0.43 | 0.451 | 0.353 | 0.389 | 0.408 | 0.44 | 0.424 | 0.432 | 0.467 | 0.509 | 0.267 | 0.304 | 0.345 | 0.395 |
| TimesFM (Zero Shot) | 0.421 | 0.472 | 0.51 | 0.514 | 0.326 | 0.399 | 0.434 | 0.451 | 0.357 | 0.411 | 0.441 | 0.507 | 0.205 | 0.294 | 0.367 | 0.473 |
| Moirai (Zero Shot) | 0.402 | 0.419 | 0.429 | 0.444 | 0.334 | 0.373 | 0.393 | 0.416 | 0.383 | 0.402 | 0.416 | 0.437 | 0.282 | 0.318 | 0.355 | 0.41 |
| Toto (Zero Shot) | 0.366 | 0.368 | 0.399 | 0.424 | 0.197 | 0.231 | 0.260 | 0.355 | 0.328 | 0.353 | 0.389 | 0.429 | 0.27 | 0.315 | 0.319 | 0.374 |
| **Auto-AR (Zero Shot)** | 0.416 | 0.464 | 0.511 | 0.53 | 0.311 | 0.416 | 0.479 | 0.555 | 0.348 | 0.426 | 0.489 | 0.854 | 0.196 | 0.272 | 0.357 | 0.549 |

| Model | Weather | | | | Electricity | | | | ILI | | | | Traffic | | | |
|---|---|---|---|---|---|---|---|---|---|---|---|---|---|---|---|---|
| Horizon | 96 | 192 | 336 | 720 | 96 | 192 | 336 | 720 | 96 | 192 | 336 | 720 | 96 | 192 | 336 | 720 |
| TEMPO (Zero Shot) | 0.254 | 0.298 | 0.332 | 0.379 | 0.276 | 0.293 | 0.309 | 0.355 | - | - | - | - | 0.343 | 0.355 | 0.356 | 0.376 |
| TimesFM (Zero Shot) | 0.122 | 0.169 | 0.242 | 0.391 | 0.119 | 0.137 | 0.158 | 0.206 | - | - | - | - | 0.327 | 0.353 | 0.378 | 0.42 |
| Moirai (Zero Shot) | 0.212 | 0.25 | 0.282 | 0.322 | 0.299 | 0.31 | 0.323 | 0.347 | - | - | - | - | - | - | - | - |
| Toto (Zero Shot) | 0.223 | 0.267 | 0.291 | 0.356 | 0.212 | 0.232 | 0.249 | 0.291 | - | - | - | - | - | - | - | - |
| **Auto-AR (Zero Shot)** | 0.184 | 0.235 | 0.3 | 0.664 | 0.158 | 0.177 | 0.205 | 0.278 | 0.987 | 0.949 | 0.936 | 0.940 | 0.461 | 0.503 | 0.557 | 0.646 |

Table 21: Zero-shot results (MAE). "-" indicates unknown quantities.

