# OpenReview forum: "Specialized Foundation Models Struggle to Beat Supervised Baselines"
_ICLR.cc/2025/Conference — ICLR 2025 Poster_

### Official Review · Reviewer_9w92 · 2024-10-28

**Soundness:** 2
**Presentation:** 3
**Contribution:** 2
**Rating:** 6
**Confidence:** 3

**Summary:**

The paper conducts a comparitive study of foundation models across three domains of time-series data, genomics and satellite data, whereby a selection of foundation models in each domain are compared to a supervised baseline. More specifically, the authors propose an Auto-ML pipeline based on DASH (Shen et al. 2022)and ASHA (Li et al. 2020). The results show that foundation models are not
able to consistently beat supervised tuned baselines across these domains.

**Strengths:**

Foundation models are a hot topic in various machine learning research domains, and it is important to have appropriate evaluation regimes that can demonstrate the efficacy or show weaknesses in developed models and test whether they are able to deliver on their promises. Therefore, the research question posed by the authors is highly relevant and of importance to the community. Similarly, it is good that the authors focus on a rather straightforward simple baseline tuning technique that can be employed across domains.

**Weaknesses:**

I do understand the authors' idea and motivation to show subpar performance of foundation models against a well tuned supervised baseline across different domains. However, by attempting to do this across quiet different domains, I believe that the paper and methodology is falling short of the specific nuances of each of the included subdomains of genomics, time-series and satellite data.

I will mainly attempt to make this point for the satellite data domain as this is the most closely related one to my background among the three:
- Earth Observation (EO) data encompasses a wide array of sensors and modalities. Additionally, there are a diverse set of tasks, classification, segmentation and pixel-wise regression just to name a few for optical data. Foundation models aim to perform well across these tasks based on a common encoder that yields relevant representations that can then be "decoded" for the task at hand. However, while you do select several classification datasets, the evaluation is limited to classification and does not include segmentation which is arguably more relevant in EO. Therefore, I would argue it is difficult to make your overall claim, if you do not compare across EO tasks, for example including segmentation, as this would align more with the premise of FMs.
- In line 352 you say "we only consider a restricted subset of top-performing, open-source, and compatibly-formatted models". However, I would believe that there are other relevant and easily accessible recent models available like [Clay](https://github.com/Clay-foundation/model), [Scale-MAE](https://github.com/microsoft/torchgeo/blob/main/torchgeo/models/scale_mae.py), the [Satlas-model](https://github.com/microsoft/torchgeo/blob/main/torchgeo/models/swin.py), or [DOFA](https://github.com/microsoft/torchgeo/blob/main/torchgeo/models/dofa.py) that include evaluations against your chosen models of CROMA and Sat-MAE.
- Regarding the experimental findings, I would argue that this is not necessarily new, because there are works that already show strong performances of supervised baselines. For example in the [Prithvi by Jakubik et al.](https://arxiv.org/abs/2310.18660) Table 4 you observe the competitive UNet performanceand that for subclasses it is often better than the FM. Similarly, in [DOFA by Xiong et al.](https://arxiv.org/abs/2403.15356), Table 1 and 2 show that fully trained supervised baselines can sometimes beat the finetuning schemes. More general a work by [Corley et al.](https://openaccess.thecvf.com/content/CVPR2024W/PBVS/html/Corley_Revisiting_Pre-trained_Remote_Sensing_Model_Benchmarks_Resizing_and_Normalization_Matters_CVPRW_2024_paper.html) conducted a focused study around pretrained weights and demonstrated that carefully finetuned baselines are very competitive.
- From Table 6 it seems that you have conducted the experiments on quiet large datasets in the domain. However, the more interesting application of FMs is for fine tuning on datasets with scarce/few labels where training a model from scratch would likely just lead to over fitting. This domain relevant scenario is not considered here.

Given that for the satellite subdomain only a subset of existing FMs were used for classification (which is understandable since this requires significant experimental effort), I think some of the author's general statements do not follow the methodology. For example, the authors lead their discussion by stating in line 470 that "At a high level, our results show that the foundation models in these three domains have not yet surpassed supervised learning" but the selected methods are not representative of "the foundation models in the domain as illustrated for satellite data. As I said in the beginning, I believe that there are too many nuances within the separate domains that can be all included in one paper and allow for the conclusions that the authors draw.

**Questions:**

General Questions:
- I don't really understand the importance of including a ResNet and a UNet, as models for classifcation. From my understanding a UNet is an architecture choice where you need the model output to also be a HxW similar to the input. For classification this would imply that you have to include additional layers to get from that output to a vector-valued classification prediction. Additionally, I think a UNet can have all sorts of different convolutional or transformer encoders, including a ResNet, for example see [here](https://github.com/qubvel-org/segmentation_models.pytorch). Having access to the code would probably answer this question.
- In line 255, you state that beyond a subset of the GeoBench classification tasks you also include BigEarthNet, and EuroSAT, as well as two additional ones. However, BigEarthNet and EuroSAT are already included in GeoBench
- CROMA takes as input an optical and/or sar image as the code suggests [here](https://github.com/antofuller/CROMA/blob/59505a6bcadbf36ba20767270154bf9f3067c5e7/use_croma.py#L90). I wonder how that is handled across the selected different classification datasets, as it was not entirely clear to me from the appendix around line 953
- Is it possible to look at the code already which could help answer some of these questions?

Other Comments:
- in line 261 you state that "satellite imaging closely resembles RGB imaging" but that is only true for some optical satellites. Radar Satellites as well as other images are very different than RGB

Minor Comments:
- line 525 there is a typo in the word "work"
- I don't find the acronym "NAS" to be introduced. Also in line 191 you say "neural architecture search" where you use NAS beforehand already

---

> ### Author Response · Authors · 2024-11-21
> **Author Response to Reviewer 9w92 (1/3)**
>
> Thank you for your constructive review. While your specific feedback with respect to satellite data is quite valuable, we respectfully disagree with the broader criticism that *”there are too many nuances within the separate domains that can be all included in one paper and allow for the conclusions that the authors draw.”* We address this issue in detail in our [general response](https://openreview.net/forum?id=JYTQ6ELUVO&noteId=wNZGZUiB3A) (**Clarification 1**), where we note that performance on standardized benchmarks is the main way of comparing methods in ML and that evaluating on a large subset of the tasks used by developers of specialized FMs themselves to claim success is sufficient to establish our own main claim. If these tasks do not address relevant settings then our claim is still true but our results imply that better benchmarks need to be developed and agreed upon in the relevant communities.
>
> We address your specific weaknesses and questions in two separate comments below.

---

> ### Author Response · Authors · 2024-11-21
> **Author Response to Reviewer 9w92 (2/3)**
>
> ## Weaknesses
>
> 1. [*”Earth Observation (EO) data encompasses a wide array of sensors and modalities. Additionally, there are a diverse set of tasks, classification, segmentation and pixel-wise regression just to name a few for optical data. [...] I would argue it is difficult to make your overall claim, if you do not compare across EO tasks, for example including segmentation, as this would align more with the premise of FMs.*”]
> Many satellite FMs do not handle all data formats, e.g. SatMAE is built for multispectral Sentinel2 data while Scale-MAE and Cross-scale-MAE are pretrained on RGB. This leads to a lack of unified benchmarks, which we have made a best-effort attempt to handle by focusing on a widely used format (Sentinel2). Furthermore, most satellite FMs highlight their classification result (e.g. it is the first results table in SatMAE, CROMA, DOFA, etc.), so we believe the task is reasonable to focus on. While our results do not imply anything about performance on tasks like segmentation, the fact that FMs are not able to convincingly beat supervised baselines on a collection of tasks that their own authors have decided are important is strong evidence for our claim.
> 2. [*”there are other relevant and easily accessible recent models available like Clay, Scale-MAE, the Satlas-model, or DOFA”*]
> Thank you for these suggestions. Because of the lack of consistent evaluations we fine-tuned all FMs ourselves and so were constrained in how many comparisons we could make. Following your suggestion, we evaluated DOFA and Satlas on GEO-Bench (Clay / Scale-MAE have more challenging formatting / data-loaders but we will consider them in a revision, along with the rest of the tasks). As shown below, the best of these new FMs only beats DASHA by about 1.3% on average, which while better than CROMA / SatMAE is still not a convincing demonstration of the utility of large-scale pretraining (for comparison, BERT beats all supervised baselines on GLUE by about 16.8% on average).
> |Model|m-bigearthnet|m-brickkiln|m-forestnet|m-pv4ger|m-so2sat|EuroSAT|avg|
> |-|-|-|-|-|-|-|-|
> |DOFA-base|73.46|98.6|55.6|98.3|61.36|99.2|81.09|
> |DOFA-large|73.91|98.7|57.4|98.3|61.92|99.19|81.57|
> |Satlas|71.82 |98.80|52.75|98.2|57.51|99.17|79.68|
> |DASHA|72.72|98.92| 57.24|97.4|56.28|99.07|80.27|
> 3. [*”I would argue that this is not necessarily new, because there are works that already show strong performances of supervised baselines. For example in the Prithvi by Jakubik et al. Table 4 you observe the competitive UNet performanceand that for subclasses it is often better than the FM. Similarly, in DOFA by Xiong et al., Table 1 and 2 show that fully trained supervised baselines can sometimes beat the finetuning schemes. More general a work by Corley et al. conducted a focused study around pretrained weights and demonstrated that carefully finetuned baselines are very competitive.”*]
> As discussed in the following, we do not believe these works undermine our novelty:
> * Jakubik et al. show that a supervised UNet is competitive with a single FM on a single non-standard task. In contrast, our claims rely on evidence that supervised methods are competitive with *multiple* FMs across *multiple* standard tasks (and across *multiple* different domains). Their result is only weak evidence for our claim (which they do not make) and so does not affect its novelty.
> * Xiong et al. show that supervised models are competitive with a linearly probed FM, whereas we show that supervised models are competitive with a *fully fine-tuned* FM. Our comparison is with a much stronger method that better reflects what FM practitioners do (given enough compute) when aiming to maximize accuracy.
> * Corley et al. show that ImageNet-pretraining is a strong baseline for satellite imagery when properly tailored. We categorize FMs pretrained on out-of-modality data but tailored to in-modality tasks as specialized FMs (c.f. SwinT-base in Table 2 and GPT4TS (OFA) in Table 3) and so we view that work as comparing between specialized FMs. Their unpretrained models are *not* competitive with the FMs they study.
> 4. [*”the more interesting application of FMs is for fine tuning on datasets with scarce/few labels where training a model from scratch would likely just lead to over fitting*”]
> We address concerns regarding data-scarce settings in the [general response](https://openreview.net/forum?id=JYTQ6ELUVO&noteId=bhK1QAbmtL) (**Limited data**). Notably, we conduct experiments that show that even when both are provided much less data, supervised baselines are competitive with specialized FMs (for satellite the drop in performance is roughly the same). Furthermore, while we view this as an interesting question, we believe experiments on benchmarks chosen by the creators of specialized FMs themselves suffice to justify our claim that the same FMs have not yet been shown to outperform supervised learning.

---

> > ### Author Response · Authors · 2024-11-21
> > **Author Response to Reviewer 9w92 (3/3)**
> >
> > ## Questions
> >
> > 1. [*”I don't really understand the importance of including a ResNet and a UNet, as models for classification. From my understanding a UNet is an architecture choice where you need the model output to also be a HxW similar to the input. [...] Having access to the code would probably answer this question.”*]
> > While we acknowledge that UNet was originally designed for segmentation tasks, it can be adapted for classification tasks (e.g. we do so by adding a global average pooling layer) and may be effective due to its ability to preserve spatial features and extract multiscale features. As discussed in the Section 4.1.2 of our paper, including DASHA in UNet consistently outperformed WRN on several genomics tasks. Lastly, note that we uploaded code along with our paper submission. Please refer to the Dashed-Dreams/DASH/src/networks directory for a detailed implementation.
> > 2. [*”In line 255, you state that beyond a subset of the GeoBench classification tasks you also include BigEarthNet, and EuroSAT, as well as two additional ones. However, BigEarthNet and EuroSAT are already included in GeoBench.”*]
> > Thank you for pointing this out and our apologies for any confusion. The original BigEarthNet dataset evaluated by other foundation models includes 19 labels. However, the GeoBench version of BigEarthNet uses a different labeling scheme and includes 43 labels, along with a more balanced label distribution. The differences between the two datasets are evident from the mAP results, which justify treating them as distinct datasets in our evaluation. On the other hand, we excluded the GeoBench version of EuroSAT from our evaluation due to its high similarity to the original EuroSAT dataset. To ensure comparability with prior work evaluating foundation models, we opted to use the original version of EuroSAT. We will make these details clear in the revision.
> > 3. [*”CROMA takes as input an optical and/or sar image as the code suggests here. I wonder how that is handled across the selected different classification datasets, as it was not entirely clear to me from the appendix around line 953”*]
> > Most of the datasets we examine consist of optical images. Therefore, following the instructions in the CROMA paper, we process these datasets through the optical encoder. If necessary, the input data is padded to ensure compatibility, and the resulting optical representation is passed through a pooling layer and a linear layer for classification. The exception is the m-so2sat dataset, which includes both Sentinel-1 (radar) and Sentinel-2 (optical) channels. For this dataset, we compute the CROMA radar encoder inputs using the first four Sentinel-1 channels (VV and VH real & imaginary) and process the optical channels through the CROMA optical encoder. The outputs from both encoders yield the multi-modal representation, which is subsequently used for classification. We will add these details in revision.
> > 4. [*”Is it possible to look at the code already which could help answer some of these questions?”*]
> > Please note that code was included along with our submission as supplementary material.
> > 5. [Comments]
> > Thank you for noting the additional writing issues, which we will correct in a revised version. Note that we meant to place the statement about RGB imaging more clearly in the context of many practitioners applying ImageNet-pretrained models to this domain.

---

> ### Author Response · Authors · 2024-11-25
> **Follow-up to Reviewer 9w92**
>
> We’re reaching out to follow up on our rebuttal and to check if you have any further questions. In our response, we addressed your concerns about domain-specific nuances and argued for the soundness of using existing benchmarks to justify our main claim. Following your suggestions, we also conducted new experiments comparing our baselines to additional satellite foundation models and assessing them in low-data settings; we believe the results we reported further validate our main claim. We hope that these responses resolve the weaknesses outlined in your review and would appreciate a prompt response to confirm that or with further questions / clarifications. Thank you again for your thoughtful feedback!

---

> > ### Comment · Reviewer_9w92 · 2024-11-26
> > **Rebuttal Answer**
> >
> > I thank the reviewers for the detailed response.
> >
> > 1. I believe my argument still stands, because to me you are making a different argument. If one has a pretrained EO foundation model whose embeddings are useful for relevant downstream tasks, let's say EO classification and segmentation, then stating that a method beats that EO foundation model on classification but also a bunch of other unrelated tasks like univariate time-series implies that this method is superior than the EO FM is not as strong to me as showing that it is superior to a series of tasks that a specific FM model was developed for since the whole premise of FMs is good performing finetuning on a series of domain relevant tasks.
> >
> > 2. Thank you for the additional effort you have put in for these experiments.
> >
> > 3. Regarding your comment about UNet, while it is certainly possible to change a UNet architecture for classification, I am still not convinced that it is a relevant classification baseline.
> >
> > Overall, I view the proposed methodology as an interesting approach and appreciate the author's detailed response. While I do not think that the author's conclusion follow their experimental setup as clearly as they claim, I still believe that the author's proposed methodology is relevant for the different domains and have updated my score accordingly.

---

> > > ### Author Response · Authors · 2024-11-27
> > >
> > > Thank you for taking the time to read through our response and providing positive feedback.
> > >
> > > With respect to your first point: we agree that outperforming EO FMs on other EO tasks would make our claim stronger *if* we did not have to choose (due to compute / page limitations) between that and showing the same thing for genomics / time series FMs on genomics / time series tasks. Since our claim concerns multiple domains, we believe outperforming domain-specific FMs on multiple domains—but studying only one type of task from each—is stronger than choosing one domain but outperforming FMs on multiple types of tasks.
> > >
> > > With respect to your third point: we agree that UNet is a strange choice for classification. It is only included because it is in the architecture search space used by DASHA and so its performance demonstrates that the method is doing something nontrivial. Thanks again for your thoughtful suggestions and feedback.

---

### Official Review · Reviewer_hQrf · 2024-10-29

**Soundness:** 4
**Presentation:** 3
**Contribution:** 3
**Rating:** 6
**Confidence:** 4

**Summary:**

This work evaluates the effectiveness of foundation models (FMs) in genomics, satellite data, and time series, finding that well-tuned supervised models can match or outperform FMs in these specialized domains. The results suggest that large-scale pretraining benefits are not yet fully realized in these areas, highlighting the need for solid baseline comparisons.

**Strengths:**

1) The paper addresses a highly relevant topic that warrants investigation, given the growing popularity of FMs and the ongoing efforts within the community to adapt these models to specialized domains beyond natural images and text.

2) The results are surprising and critical to keep a fair benchmarking of future FMs against other pipelines. The actual gains in data efficiency scored by the proposed baselines (nice summary in Figure 1) range from three to five orders of magnitude in the three tested domains (satellite imaging, genomics, and time series). The authors also plan to release the code for their baselines, providing the community with a valuable benchmark resource.

3) The experimental setup is comprehensive, with models evaluated on up to 50 tasks. I found the design of the supervised baseline to be rigorous and fair, especially the decision to approximate standard practitioner model development through architecture search.

4) The paper is clear, well-written, and easy to follow, making its main message easy for the reader to understand.

**Weaknesses:**

1) The paper presents two "solutions" from an architectural standpoint, i.e., CNNs for genomics/satellite data and auto-AR for time series. This leads to two questions:
- What model should a practitioner use when facing a task in a different specialized domain?
- How come 1D CNNs (or, e.g., RNNs) fail to match the performance of FMs in time series forecasting?

Is it just a matter of domain knowledge and data type? In practice, the paper would benefit from some more convincing justification and evidence for the statement, "While DASHA can be applied to forecasting tasks, it is not competitive with state-of-the-art time series FMs." (Line 240).
Also, why in Table 3 the evaluation of DASHA for the "Full setting" panel is reported as "unknown" ("-")?

2) The authors should add a brief discussion either in the Introduction or Related Work with references regarding "simple tuned baselines", which were shown to outperform/match more complex approaches, as this fact has been seen multiple times in the machine/deep learning literature and is clearly relevant to this work. Some related literature:

- Evaluating Weakly Supervised Object Localization Methods Right, CVPR 2020
- In Search of Lost Domain Generalization, ICLR 2021
- Image Classification with Small Datasets: Overview and Benchmark, IEEE Access 2022
- OoD-Bench: Quantifying and Understanding Two Dimensions of Out-of-Distribution Generalization, CVPR 2022
- When Do Neural Nets Outperform Boosted Trees on Tabular Data?, NeurIPS 2023 Datasets & Benchmarks
- Systematic Comparison of Semi-supervised and Self-supervised Learning for Medical Image Classification, CVPR 2024

**Questions:**

See the weakness section for possible improvements. I am willing to listen to the author's feedback and raise my score during the rebuttal period if the weaknesses are addressed.

---

> ### Author Response · Authors · 2024-11-21
> **Author Response to Reviewer hQrf**
>
> Thank you for your positive response and useful feedback. We are glad that you found our paper relevant, easy to understand, and comprehensive, and we hope to address your concerns below:
>
> 1. [*“The paper presents two "solutions" from an architectural standpoint, i.e., CNNs for genomics/satellite data and auto-AR for time series.This leads to two questions: What model should a practitioner use when facing a task in a different specialized domain? How come 1D CNNs (or, e.g., RNNs) fail to match the performance of FMs in time series forecasting? Is it just a matter of domain knowledge and data type?”*]
> Thank you for the interesting questions, which we are happy to address in some detail. However, we first note that, as discussed in the [general response](https://openreview.net/forum?id=JYTQ6ELUVO&noteId=wNZGZUiB3A) (**Clarification 3**), the goal of our paper was *not* to promote tuned CNNs and Auto-AR as general-purpose multi-domain solutions, but rather to use them to show that specialized FMs have not yet outperformed supervised learning in these domains. Thus showing that any one baseline is competitive with state-of-the-art specialized FMs is sufficient to make our claims.
> * To address your first question: DASHA, our CNN-based automatic search workflow, is the more general approach and can be generalized to many other tasks, as it can work on any data type that can be processed by a CNN. The results in our **Empirical Results** section also show that DASHA achieves significant improvement upon the vanilla CNN architectures. Therefore, if people want to use our findings in the paper for stronger benchmarking in other specialized domains, we recommend DASHA. While Auto-AR is a very simple forecasting baseline, it is designed for auto-regressive prediction and so it is unclear if it can be useful even for other 1D tasks.
> * To address your second question: we agree that the underperformance of CNNs is an interesting question and one worth exploring further, although the strong performance of Auto-AR is sufficient for the purposes of the claims in our paper. Roughly speaking, the underperformance of CNNs (and possibly of RNNs) may be due to a smaller effective training data size when applied to forecasting datasets. This is because these datasets consist of overlapping points that are chunks of one single sequence, so CNNs will compute very similar representations for many of the data points due to translation invariance. Indeed we’ve found empirically that training CNNs on *fewer* chunks with *less* overlap significantly *improves* performance, despite involving fewer actual data points. Thus it may be possible to improve the performance of traditional neural networks via smarter data processing, which is an interesting direction for future research. However, the primary focus of our paper is to demonstrate that specialized FMs have not convincingly outperformed supervised baselines, so performing a more detailed investigation shifts the focus away from our main objective.
> * [*”In practice, the paper would benefit from some more convincing justification and evidence for the statement, "While DASHA can be applied to forecasting tasks, it is not competitive with state-of-the-art time series FMs." (Line 240).”*]
> We are somewhat unsure about the specific concern here, as this claim is negative about our own method. It is justified by the reported performance in Table 3.
> * [*”why in Table 3 the evaluation of DASHA for the "Full setting" panel is reported as "unknown" ("-")?”*]
>  The DASHA results in Table 3 are incomplete because the computational cost of running it on one of the tasks was prohibitively large due to the large number (862) of output channels. However, as discussed above the Auto-AR results on their own suffice for our claims related to time series FMs, and we provide the (partial) DASHA results only for additional context.
> 2. [*“The authors should add a brief discussion either in the Introduction or Related Work with references regarding simple tuned baselines”*]
> Thank you for highlighting these relevant sources. We agree that referencing these papers (as well as similar work in [text classification](https://aclanthology.org/P12-2018/), [text representation](https://openreview.net/forum?id=SyK00v5xx), [RL](https://arxiv.org/abs/1803.07055), and [architecture search](https://proceedings.mlr.press/v115/li20c.html)) is worthwhile and will include them in a revised version of the paper.

---

> ### Author Response · Authors · 2024-11-25
> **Follow-up to Reviewer hQrf**
>
> We’re reaching out to follow up on our response and to check if you have any further questions. In our rebuttal, we addressed your concerns about the choices faced by practitioners and the need for additional references about past work on baselines in ML; for the first point we also discussed and provided some empirical insights into your interesting question about the underperformance of CNNs. We hope that these responses resolve the weaknesses outlined in your review and would appreciate a prompt response to confirm that or with additional questions / clarifications. Thank you again for your thoughtful feedback!

---

> ### Comment · Reviewer_hQrf · 2024-11-25
>
> I thank the authors for the detailed rebuttal. I still have some doubts regarding the proposed solution, which is split into different approaches.
>
> **If** "showing that any one baseline is competitive with state-of-the-art specialized FMs is sufficient to make our claims" what is then the purpose of proposing an automated baseline and another approach such as AutoAR if DLinear, which is already a supervised baseline, scores an average RMSE close to AutoAR and FMs on the partial experimental setting (Table 3)?
> I still think that the actual implementation of the supervised baseline to beat FMs does matter, and indeed I consider the automated DASHA a contribution that could strengthen future FMs benchmarks.
>
> I appreciate that the authors agreed with the related literature and found additional works in other ML fields that are related to their paper. The addition of such works would definitely strengthen the paper.

---

> > ### Author Response · Authors · 2024-11-25
> >
> > Thank you for clarifying your concern. In this study we sought out the simplest competitive baselines in each domain, where “competitive” is roughly defined as “nearly matching or outperforming all FMs on one or more aggregate metrics.” This led to the same approach (DASHA) for genomics and satellite imaging but a different one (Auto-AR) for time series. While we agree that DASHA is a useful contribution for the reason you state—it can be used to strengthen future benchmarks in other domains—it is not universally applicable (since CNNs are not) and will not always do well (since CNNs do not). On time series specifically it performs about as well as DLinear, whose performance is decent but *not* sufficient to make our claim for the following reasons:
> >
> > 1. Average RMSE is just one of the four aggregate metrics we consider, and even there DLinear is non-trivially worse than Auto-AR and the leading open-source FMs. The gap between DLinear (0.551) and Auto-AR (0.534) is roughly 3%, whereas the gap between Auto-AR and the two leading open-source FMs (GPT4TS (0.540) and MOMENT (0.534)) is closer to 1%. We believe it is debatable whether DLinear is competitive with leading open-source FMs according to this metric, whereas the competitiveness of Auto-AR is **not** debatable. Being able to say that makes our overall claim much stronger.
> > 2. Every metric has weaknesses (e.g. average RMSE is sensitive to outliers), so it is important to compare across multiple metrics. Among *non* zero-shot FMs, DLinear is only ever better than just one of them on just one metric, whereas Auto-AR is as good or better than *all* open-source FMs according to three of the four metrics. We believe that matching or outperforming FMs according to multiple metrics is much more incontrovertible evidence for our claim than getting somewhat close.
> > 3. The above arguments hold similarly in the “full” (8-dataset) setting, where Auto-AR is the best according to two of the four metrics whereas DLinear is not the best on any.
> > 4. As you mention, the implementation of the baselines matters, but we believe this point favors Auto-AR, as it is an older and in many ways much simpler method than DLinear (and has a thousand times fewer parameters). Auto-AR is also simpler and more efficient than DASHA, which is another reason we chose the former.
> >
> > Lastly, we believe Auto-AR itself is *also* a valuable contribution to the time series community specifically. Indeed, the fact that a very old model with very few parameters can be made so competitive by just increasing its lookback window is perhaps the most surprising secondary result of our study. Before this, the closest thing to a baseline in this area was Auto-ARIMA (Hyndman & Khandakar, JSS 2008), which is quite popular (5000+ citations) but performs poorly on these tasks, so being able to explain its underperformance is very important and should greatly impact forecasting benchmarks. See Section 5.4 for a discussion of this.
> >
> > Thank you again for clarifying your view. We hope the above explanation, which we plan to include in revision to improve the clarity of our work, helps explain our choice of baselines.

---

> > > ### Comment · Reviewer_hQrf · 2024-11-26
> > >
> > > I would like to thank the authors again for the additional detailed explanations provided.
> > >
> > > I would like to summarize my concern, which exactly resides in the two sentences stated by the authors:
> > > - "On time series specifically DASHA performs about as well as DLinear, whose performance is decent but not sufficient to make our claim for the following reasons"
> > > - "As you mention, the implementation of the baselines matters"
> > >
> > > Aligning with my initial concern in the review, I think that the implementation of the supervised baseline is a fundamental point that practitioners should face when facing another specialized domain. I do not doubt that specialized baselines matching FMs exist since the authors have interestingly shown that. However, the practical difficulty, due to the variation of the solution, for valid problem-related reasons, makes the "finding" of the solution to support the claim "supervised baseline match FMs" harder to always be true in practice.

---

> > > > ### Author Response · Authors · 2024-11-27
> > > >
> > > > Thank you again for your positive feedback on our paper. We agree that a more general-purpose solution would make it much easier to verify the consistency of our claims, or at least to more consistently benchmark specialized FMs. Devising such a method is valuable future work, although we also believe that focusing on generality may lead to overlooking domain-specific discoveries such as the one we made with Auto-AR. In any case, for now we view the fact that our claim holds across three of the more popular specialized domains (in terms of FM development) to be a strong indicator that such issues may arise more broadly.

---

### Official Review · Reviewer_TNS8 · 2024-11-04

**Soundness:** 3
**Presentation:** 2
**Contribution:** 2
**Rating:** 6
**Confidence:** 4

**Summary:**

The paper examines the effectiveness of foundation models (FMs) in specialized domains like genomics, satellite imaging, and time series data, relative to traditional supervised learning models. The authors aim to challenge the assumption that foundation models necessarily outperform supervised models in these domains. Using a rigorous experimental framework, they contrast FMs with supervised baselines, emphasizing cases where the benefits of large-scale pretraining do not translate into superior downstream performance. Through their novel automated pipelines—DASHA for genomics and satellite imaging, and Auto-AR for time series—the authors provide a streamlined, data-efficient, and computationally lighter alternative to FMs, that demonstrates competitive performance across several tasks. The study raises important considerations about the necessity and cost-efficiency of pretraining-heavy foundation models in specialized fields.

**Strengths:**

1. The benchmarks and datasets chosen for each domain appear to comprehensively cover relevant applications in genomics, satellite imaging, and time series.

2. The paper highlights computational efficiency as a priority by contrasting the costs associated with FMs versus the supervised baselines. Holds high relevance given the fast adoption of large, resource-intensive models in research and industry.

3. Contextualized the results within each domain, providing a nuanced understanding of the model performance across tasks.

**Weaknesses:**

1. The paper lacks an exploration of their applicability to other types of data or more complex, multi-modal tasks. A broader discussion of potential limitations could help future researchers understand the contexts in which DASHA and Auto-AR might or might not succeed.

2. The choice of baselines (e.g., Wide ResNet and UNet for genomics and satellite imaging) is reasonable but could be expanded. Although the selected baselines demonstrate competitive performance, further comparisons with more recent architectures could reinforce the study’s claims regarding the relative inefficacy of FMs in these domains.

3. Although the paper effectively highlights the efficiency of supervised workflows, the focus on computational cost may not fully account for the trade-offs in scalability and generalization that FMs can offer.

4. For time series, the paper does not sufficiently address the unique challenges of zero-shot forecasting. Including more details on why certain zero-shot FMs performed poorly could provide actionable insights for model improvement in zero-shot contexts.

**Questions:**

Q1. Could the DASHA and Auto-AR workflows be extended or adapted for multi-modal datasets or tasks involving mixed data types? If so, what modifications might be necessary?
Q2. How do the authors envision the role of foundation models evolving in these domains if computational resources continue to scale? Would certain configurations of FMs potentially start to outperform supervised baselines, or do they anticipate the observed trends to persist?
Q3. Given the success of NAS in selecting optimal CNN architectures for genomics and satellite imaging, do the authors plan to explore whether reinforcement learning or meta-learning approaches might yield further performance gains?

---

> ### Author Response · Authors · 2024-11-21
> **Author Response to Reviewer TSN8**
>
> Thank you for your constructive feedback. We are glad that you found our results comprehensive and relevant, and we hope to address your concerns below.
>
> ## Weaknesses
>
> 1. [*"The paper lacks an exploration of their applicability to other types of data or more complex, multi-modal tasks."*]
> As noted in the [general response](https://openreview.net/forum?id=JYTQ6ELUVO&noteId=wNZGZUiB3A) (**Clarification 3**), the goal of our paper was *not* to promote DASHA and Auto-AR as general-purpose multi-domain solutions, but rather to use them to show that specialized FMs have not yet outperformed supervised learning in these domains. We do not make a claim that DASHA and/or Auto-AR are applicable to any tasks beyond those studied in the paper, and so investigating that is out-of-scope.
> 2. [*"The choice of baselines (e.g., Wide ResNet and UNet for genomics and satellite imaging) is reasonable but could be expanded."*]
> The central claim of our paper is that specialized foundation models struggle to beat supervised baselines, so it is sufficient to provide just one supervised approach that is competitive with FMs, which we have done for all domains under consideration. While additional baselines may add context to the results, as you mentioned this would mainly serve to reinforce the claims we already show.
> 3. [*“Although the paper effectively highlights the efficiency of supervised workflows, the focus on computational cost may not fully account for the trade-offs in scalability and generalization that FMs can offer.”*]
> While we are not sure exactly what tradeoffs this is referring to, you may be interested in the new data-scarcity experiments we report in the [general response](https://openreview.net/forum?id=JYTQ6ELUVO&noteId=bhK1QAbmtL) (**Limited data**).
> 4. [*“For time series, the paper does not sufficiently address the unique challenges of zero-shot forecasting …”*]
> As discussed in the [general response](https://openreview.net/forum?id=JYTQ6ELUVO&noteId=wNZGZUiB3A) (**Clarification 2**), our goal was to identify a broad trend across multiple domains, and so due to computational and space considerations we could not do very fine-grained investigations of specific subsets of FMs.
>
> ## Questions
> 1. [*”Could the DASHA and Auto-AR workflows be extended or adapted for multi-modal datasets or tasks involving mixed data types? If so, what modifications might be necessary?”*]
> While Auto-AR is an autoregressive method that is specifically suited for time series forecasting, DASHA is likely much more suitable for multi-modal or mixed-type settings; in fact its architecture search component ([DASH](https://arxiv.org/abs/2204.07554)) was originally developed and evaluated on a very diverse set of domains. For multi-modal / mixed-type settings it is likely that DASHA would need to allow for multiple parallel backbones to process the different inputs modalities / types, whereas the code currently works with one backbone at a time.
> 2. [*”How do the authors envision the role of foundation models evolving in these domains if computational resources continue to scale? Would certain configurations of FMs potentially start to outperform supervised baselines, or do they anticipate the observed trends to persist?”*]
> As discussed in our work, FMs in NLP have dominated traditional supervised approaches since the advent of large-scale pretraining. While similar performance gains have not yet been observed in these specialized domains, we think that the availability of compute and large-scale data suggests that with better pretraining approaches and perhaps modeling they will eventually also see (specialized) FMs that outperform supervised learning. Our study highlights the need for better benchmarks and baselines for driving this progress.
> 3. [*”Given the success of NAS in selecting optimal CNN architectures for genomics and satellite imaging, do the authors plan to explore whether reinforcement learning or meta-learning approaches might yield further performance gains?”*]
> While the focus of our work was simply to establish the existence of supervised baselines that outperform specialized FMs, we agree that the success of our NAS-based approach suggests that data-driven model development (including as you mention RL and meta-learning) may yield strong performance gains, both for supervised methods and transfer-based approaches. Research in this direction is indeed a likely fruitful area for future work.

---

> ### Author Response · Authors · 2024-11-25
> **Follow-up to Reviewer TNS8**
>
> We’re reaching out to follow up on our response and to check if you have any further questions. In our rebuttal, we addressed your four concerns about the scope of our paper and the resulting choice of experiments to run. We hope that this resolves the weaknesses outlined in your review and would appreciate a prompt response to confirm that or with additional questions / clarifications. Thank you again for your thoughtful feedback!

---

> > ### Comment · Reviewer_TNS8 · 2024-12-01
> > **Official Response to Authors**
> >
> > Thank you for the response. It provides useful clarifications on certain aspects of the study.

---

### Official Review · Reviewer_E1xK · 2024-11-04

**Soundness:** 3
**Presentation:** 3
**Contribution:** 3
**Rating:** 8
**Confidence:** 5

**Summary:**

This paper aims to challenge the assumption that foundation models (FMs) consistently outperform traditional supervised learning methods. To support this, the authors design an automated machine learning framework, utilizing neural architecture search for genomics and satellite imaging tasks, and an Auto-ARIMA approach for time series analysis. These automated workflows produce competitive supervised models, providing strong baselines against which FMs are compared.

The study highlights that, unlike in general-purpose AI tasks (e.g., NLP or computer vision), the advantages of FMs in specialized fields remain unproven (FMs frequently fail to surpass traditional supervised learning models that are optimized for each task), emphasizing the need for diverse, well-tuned baselines in FM evaluation.

**Strengths:**

1. This paper is well-organized and well-written. The research questions are both interesting and practical, particularly in highlighting that some foundation models (FMs) are not fully comparable to in-domain, supervised baseline models. The experimental setup and results are comprehensive, effectively supporting the study’s objectives and providing valuable insights into the performance of FMs relative to tailored supervised models.

2. The two proposed automated supervised pipelines, DASHA and Auto-AR, are designed with detailed insights into model architecture, resulting in lightweight, simple yet powerful baselines. In particular, the development of Auto-AR highlights previously overlooked limitations and gaps in earlier works, encouraging the research community to rethink classical methods.

3. The paper demonstrates high quality through its methodological rigor and detailed experiments. (1) Thorough Benchmarking: The authors evaluate multiple foundation models (FMs) across three distinct scientific fields, using datasets and tasks that reflect real-world applications. This comprehensive experimental setup ensures that the results are robust and meaningful. (2) Well-Constructed Baselines: The paper establishes carefully optimized baselines using existing supervised models. This level of detail ensures fair and well-founded comparisons, providing a reliable foundation for drawing conclusions.

**Weaknesses:**

1. The strong performance of supervised methods through the automated pipeline may be largely attributed to the large-scale training data. As noted in the data statistics in the appendix, nearly all datasets contain over 10,000 samples for training, which is sufficient for effective supervised training from scratch, even without pretrained checkpoints. For a comprehensive evaluation and to support the claim that pure supervised learning can outperform pretraining, additional experiments varying data scale (e.g., using different ratios of training data) are needed. This would likely highlight the advantages of large-scale pretraining.

2. In addition to performance metrics like accuracy and model parameters, other essential factors are not addressed in the paper. Specifically, the authors do not mention the time and computing resources required to obtain the optimal architecture and hyperparameters. Compared to directly fine-tuning foundation models (FMs), it is unclear whether the automated pipeline is more time- and compute-intensive. Including these comparisons would provide a fuller understanding of the trade-offs involved in using the automated pipeline.

3. The assumption that “their creators make a best-effort attempt to present their own method in the best light” may not be fully reasonable. Due to the high computational costs and complexity of foundation models, obtaining optimal or even sub-optimal hyperparameters is generally challenging.

**Questions:**

My primary concerns regarding the acceptance of this work are based on points 1 and 2 described in the “weakness” section. These concerns stem from my own experience, particularly regarding point 1, where foundation models (FMs) generally demonstrate greater robustness in small-scale data adaptation through fine-tuning in vision and language tasks. However, this behavior may differ in the other modalities used in this paper.

The claim that “FMs in these domains have not yet surpassed supervised workflows and are often outperformed by fairly simple methods, including lightly modified CNN backbones (in genomics and satellite imaging) and classical linear forecasters (for time series)” may be somewhat overstated. This is especially relevant when considering factors such as data bias, the challenges of hyperparameter optimization for FMs, and potential out-of-distribution issues that could lead to negative transfer due to pretraining data limitations.

---

> ### Author Response · Authors · 2024-11-21
> **Author Response to Reviewer E1xK**
>
> Thank you for your constructive feedback. We are glad that you found our work comprehensive and detailed, and we hope that the additional experiments we discuss below and in our general response can address your concerns. Before that, please note that we address your broader concerns that our claims may be overstated due to not considering certain settings (e.g. small-scale data adaptation, data bias, negative transfer) in our [general response](https://openreview.net/forum?id=JYTQ6ELUVO&noteId=wNZGZUiB3A) (**Clarification 1**). In particular, because the benchmarks we use are exactly the ones used by developers of specialized FMs to claim success, they are sufficient to establish our main claim regardless of whether or not one believes that the standard benchmarks address the appropriate settings. If they do not then our claim is still true but better benchmarks need to be developed in the relevant communities.
>
> 1. [*"The strong performance of supervised methods through the automated pipeline may be largely attributed to the large-scale training data ..."*]
> We have addressed concerns regarding the performance of foundation models in data-scarce settings in the [general response](https://openreview.net/forum?id=JYTQ6ELUVO&noteId=bhK1QAbmtL) (**Limited data**). As explained there, while we view this as an interesting question and show some results comparing our baselines and top FMs when given limited data, we believe experiments on benchmarks chosen by the creators of specialized FMs themselves are sufficient to justify our claim that the same FMs have not yet been shown to outperform supervised learning.
> 2. [*"... it is unclear whether the automated pipeline is more time- and compute-intensive ..."*]
> We agree that this is a valid concern and have addressed it in the [general response](https://openreview.net/forum?id=JYTQ6ELUVO&noteId=bhK1QAbmtL) (**Computational cost**). As detailed there, we have compared the cost of our baselines to fine-tuning FMs on a representative subset of the tasks; our finding is that running our baselines is not unreasonably more costly and sometimes even cheaper than fine-tuning FMs on the same tasks. We will extend these experiments to all the tasks in the revision.
> 3. [*"The assumption that “their creators make a best-effort attempt to present their own method in the best light” may not be fully reasonable. Due to the high computational costs and complexity of foundation models, obtaining optimal or even sub-optimal hyperparameters is generally challenging."*]
> We acknowledge that hyperparameter optimization for specialized FMs presents a significant challenge due to their high computational costs and complexity. However, we believe that the authors of these FMs have likely made considerable efforts to tune both the architectures and hyperparameters of their models, efforts that may not be feasible for a typical practitioner looking to apply these FMs to a different downstream task. While such challenges are far less prominent with simple supervised learning baselines, we view this as an inherent limitation of current specialized FMs. Moreover, our evaluation does not involve selecting specific datasets but focuses on benchmark tasks widely used by several specialized FMs. It is important to note that these specialized FMs have already undergone extensive domain-specific pretraining with the eventual goal of improved downstream performance in these tasks. Therefore, even if hyperparameters are not well optimized, the FMs' pretraining should still reflect their ability to perform across tasks.

---

> > ### Comment · Reviewer_E1xK · 2024-11-26
> >
> > Thank you for providing additional experiments and detailed responses. Based on your rebuttal, I have the following comments:
> >
> > [Concerns about Computational Costs and Performance (Rebuttal Points (a) and (b)):]
> > The computational cost experiments (a) and limited data experiments (b) raise concerns about the claims and methodology proposed in this paper. Specifically:
> >
> > In (a), supervised methods generally require significantly more computational resources compared to fine-tuning foundation models (FMs). Despite this higher cost, their performance is relatively lower or comparable, rather than achieving substantial improvements over FMs.
> >
> > Considering both data-sufficient and data-scarce scenarios ((a) and (b)), the same phenomenon is observed: supervised training fails to surpass FM fine-tuning in performance. These results suggest that while FMs are not always state-of-the-art, they remain robust and user-friendly.
> >
> > For example, on datasets such as m-brickkiln, m-bigearthnet, and EuroSAT, the architecture search and hyperparameter selection for your proposed method take 11.96 GPU hours—nearly double the time required for directly fine-tuning FMs. This observation contradicts the claim that “our baselines are not unreasonably more costly and sometimes even cheaper than fine-tuning FMs on the same tasks.”
> >
> > [Third Response Regarding Hyperparameter Optimization:]
> > I partially agree with your statement in the third response; however, it is not a primary concern for this review. That said, I still recommend removing the corresponding statement. Many works follow prior configurations to ensure fair comparisons, rather than engaging in extensive hyperparameter optimization.
> >
> > [Summary:]
> > Based on the extensive experiments provided, I do not observe significant advantages of the proposed supervised pipeline compared to fine-tuning FMs. This paper is not primarily an evaluation paper aimed solely at highlighting FMs’ weaknesses in specific domains. As a technical paper, it should emphasize the performance and technical contributions of the proposed methodology.
> > For acceptance at a venue like ICLR, the proposed method should demonstrate either:
> > 	•	Superior efficiency with comparable performance, or
> > 	•	Superior performance with an acceptable increase in computational cost.
> > If neither efficiency nor effectiveness is demonstrated to a sufficient level, the paper does not currently meet the acceptance threshold.

---

> ### Author Response · Authors · 2024-11-25
> **Follow-up to Reviewer E1xk**
>
> We’re reaching out to follow up on our response and to check if you have any further questions. In our rebuttal, we addressed your concerns about the size of the fine-tuning data, the computational cost of our baselines, and hyperparameter optimization; for the first two we also reported new experimental results that confirm our main claim. In addition, we discussed your broader concerns about the claims in our paper in the general response. We hope that this resolves the weaknesses outlined in your review and would appreciate a prompt response to confirm that or with additional questions / clarifications. Thank you again for your thoughtful feedback!

---

> ### Author Response · Authors · 2024-11-27
>
> Thank you for taking the time to respond. We believe your concerns stem from (1) a misunderstanding of our goal and (2) a miscomparison when assessing the new cost experiments (although we acknowledge the latter should have been better presented). We hope to resolve this below.
> # (1) Our goal
> Your response states *“This paper is not primarily an evaluation paper aimed solely at highlighting FMs’ weaknesses in specific domains. As a technical paper, it should emphasize the performance and technical contributions of the proposed methodology.”* We would like to respectfully but strongly push back on this, as the **exact opposite** is true: our goal is *precisely* to highlight FMs’ weaknesses in specific domains. This is reflected in
> 1. Our title
> 2. The last sentence of our abstract
> 3. Our intro: “our study’s motivating question: *Do these new specialized FMs outperform traditional supervised learning applied to the same tasks?*”
> 4. The first sentence of our conclusion
> 5. Your [initial summary](https://openreview.net/forum?id=JYTQ6ELUVO&noteId=OJX6LvImgW): *“This paper aims to challenge the assumption that foundation models (FMs) consistently outperform traditional supervised learning methods. [...] The study highlights that, unlike in general-purpose AI tasks [...] the advantages of FMs in specialized fields remain unproven”*
> 6. Our [general response](https://openreview.net/forum?id=JYTQ6ELUVO&noteId=wNZGZUiB3A) (**Clarification 3**)
>
> We hope you will reassess the new results in light of our goal being **not** to propose methods but to evaluate FMs. In particular, to show the main claim of our paper it suffices to show supervised baselines that match FMs with a similar compute budget.
> # (2) Cost experiments
> Your response states *“supervised methods generally require significantly more computational resources compared to fine-tuning foundation models.”* This assessment comes from comparing the cost of tuning AND training our baselines to the cost of **just** fine-tuning FMs; as alluded to in our [original rebuttal](https://openreview.net/forum?id=JYTQ6ELUVO&noteId=bhK1QAbmtL) and detailed below, when *tuning* FM fine-tuning is factored in the cost of our baselines is as good or better:
> 1. In genomics, fine-tuning FMs takes roughly as long as training our baseline (~0.26 GPU-hours). While developing / tuning our models takes 1.41 GPU-hours, we looked into how the listed FMs were tuned (Section A.1.4 [here](https://www.biorxiv.org/content/10.1101/2023.01.11.523679v3)) and found that it was through 10-fold cross-validation, i.e. 10 runs on 90% of the data each. Thus the tuning costs of the FMs can be inferred to be roughly 2.34 = 10 x 0.9 x 0.26 GPU-hours, more than 1.5x the cost of developing / tuning our models.
> 2. In satellite, fine-tuning FMs takes 1.5-3 times as long as training our baselines (6.26 and 11.87 GPU-hours for the FMs vs. 3.70 for the baselines). While we do not know how long tuning took in the original evaluations, in our own evaluations we tried 4 different learning rates (Section B), leading to a total cost of 25.04 = 4 x 6.26 and 47.48 = 4 x 11.87 GPU-hours for the FMs, which is also roughly 1.5-3x as long as the total cost of our baselines (15.66 = 11.96 + 3.70 GPU-hours).
> 3. In time series, our pipeline is >30x faster than fine-tuning FMs.
>
> At the end of this response we present a new version of the tables that accurately reflects these inferred costs. In our revised version we will include these numbers as important context.
> # Summary
> Your response states *“Despite this higher cost, [the supervised baselines’] performance is relatively lower or comparable, rather than achieving substantial improvements over FMs.”* As discussed in point (2) above, the cost of our baselines is comparable or lower. Furthermore, based on Figure 1 in our paper we believe it is more accurate to say that the performance of our baselines is either relatively *higher* (genomics) or comparable (satellite, time series). This suffices to show the FMs do not outperform supervised baselines on these tasks, which as discussed in point (1) above is the goal of our paper.
>
> Thank you again for your constructive review of our paper and rebuttal, both of which will improve the presentation of the results. We hope that you will reassess your concerns in light of these clarifications and would be happy to answer any further questions.
> # Updated cost tables
> ## Genomics (GPU-hours)
> |Method|Training / fine-tuning|Development + tuning + training / fine-tuning|
> |---|---|---|
> |HyenaDNA-1K|0.260|~2.34|
> |NT-Multispecies (500M)|0.263|~2.367|
> |DASHA|0.256|1.666|
> ## Satellite (GPU-hours)
> |Method|Training / fine-tuning|Development + tuning + training / fine-tuning|
> |---|---|---|
> |CROMA-Large|11.87|~47.48|
> |SatMAE-Large|6.26|~25.04|
> |DASHA|3.70|15.66|
> ## Time series (GPU-mins)
> |Method|Training / fine-tuning|Development + tuning + training / fine-tuning|
> |---|---|---|
> |GPT4TS|41.23|≥41.23|
> |MOMENT|19.44|≥19.44|
> |Auto-AR|0.07|0.55|

---

> > ### Comment · Reviewer_E1xK · 2024-11-27
> >
> > Thank you for your quick response. I believe the updated results address my concerns, and your clarification of the paper’s goals is clear. I will adjust my previous scoring accordingly.

---

### Author Response · Authors · 2024-11-21
**General Response to Reviewers (1/2)**

We thank the reviewers for their time and thoughtful feedback. We are happy that they collectively recognized several key strengths of our paper:
1. Interesting and highly practical research questions (all)
2. Comprehensive experimental setup encompassing a wide range of tasks across 3 distinct domains (E1xK, TNS8, hQrf)
3. Rigorously designed workflow that brings attention to overlooked topics (E1xk, hQrf)

In our general response we provide some clarifications related to concerns raised by multiple reviewers and (in the comment below) some new experimental results. We plan to include these clarifications and (expanded versions of) the new experiments in a future revision.
## Clarifications
1. Some reviewers raised concerns that our claims may be overstated or hard to show due to difficulty accounting for few-shot regimes, data bias, negative transfer, domain-specific nuances, etc. Our paper’s main claim is that, in many domains, specialized foundation models (FMs) have not been convincingly shown to outperform supervised learning. To show this it suffices to evaluate on benchmarks that have been used by developers of the same FMs to claim success. This is because if the main benchmarks in a domain are *insufficient* to demonstrate that its FMs struggle to beat supervised baselines then they are *also* insufficient to demonstrate that its FMs *do* convincingly beat supervised baselines, implying our main claim. Thus, depending on one’s view of the benchmarks, our negative results may be interpreted as evidence either of the lack of good of FMs in these domains *or* that their important benchmarks fail to demonstrate the true utility of FMs (e.g. due to too much data, irrelevant tasks, etc.). In the latter case our results are still significant, as performance on agreed-upon benchmarks is the standard way of making falsifiable claims (and thus progress) in ML. Establishing benchmarks that properly address individual in-domain use-cases of FMs is a matter for the individual communities and out-of-scope of our paper.
2. Relatedly, some reviewers expressed interest in detailed studies of specific domains, e.g. few-shot regimes or different downstream tasks. The goal of our experiments is to highlight a broad trend across multiple domains while staying within computational and page limitations, so we focus on a subset of tasks that have been prominently identified within each domain as indicative of successful pretraining. They have all been used by FMs as evidence of their success, and so achieving consistently similar or better performance with supervised methods is enough to establish our claim.
3. Lastly, some reviewers expressed interest in the broader applicability of our baselines. We would like to clarify that the primary goal of our study is to investigate whether specialized FMs have outperformed supervised baselines on the former’s own benchmarks. To do so we develop new automated baselines, but the fact that they match or outperform those FMs is sufficient to support our claims. Simplifying and automating the application of our baselines is a secondary goal.

---

> ### Author Response · Authors · 2024-11-21
> **General Response to Reviewers (2/2)**
>
> # New experiments
>
> We provide new experimental results to address some specific concerns raised by reviewers. Due to time constraints we conduct these experiments only on a subset of tasks, but we will include numbers for all tasks in each domain in the final version.
> ## Computational cost experiments
> While we used dataset size and parameter counts as measures of cost, Rev. E1xK raised the valid question of whether our “automated pipeline is more time- and compute-intensive.” To address this, in each domain we take a representative subset of tasks and compare the automated model development and training costs of our baselines to the costs of fine-tuning the top two FMs (i.e. those in Fig. 2). As shown below, for satellite / time series our *full* pipelines (including tuning) are at worst only slightly slower than fine-tuning the best FM (*excluding* tuning). For genomics, our training costs are roughly the same as FM fine-tuning costs while our architecture search + hyperparameter tuning costs are roughly 5x more expensive than FM *fine-tuning* costs. While we do not know how long hyperparameter tuning took for the genomics FMs, we believe the equivalent of trying five different learning rates is a reasonable computational budget. All experiments were run on L40 (L40S for genomics) GPUs and will be extended to all tasks in the revision.
> #### Avg. genomics costs (GPU-hours) across 4 tasks: enhancers, H3, promoter_all_ splice_sites_all
> | Method | Model development + tuning | Training / fine-tuning |
> |---|---|---|
> | HyenaDNA-1K  | -- | 0.260 |
> | NT-Multispecies (500M)  | -- | 0.263 |
> | DASHA | 1.410 | 0.256 |
> #### Avg. satellite costs (GPU-hours) across 3 tasks: m-brickkiln, m-bigearthnet, and EuroSAT
> | Method | Model development + tuning | Training / fine-tuning |
> |---|---|---|
> | CROMA-Large | -- | 11.87 |
> | SatMAE-Large | --  | 6.26  |
> | DASHA | 11.96  | 3.70 |
> #### Avg. time series costs (GPU-mins) across 4 horizons of 2 tasks: ETTh1 and Weather
> | Method | Model development + tuning | Training / fine-tuning |
> |---|---|---|
> | GPT4TS  | -- | 41.23 |
> | MOMENT | -- | 19.44 |
> | Auto-AR | 0.48 | 0.07 |
>
> ## Limited data experiments
>
> Revs. E1xK and 9w92 suggest that the *“strong performance of supervised methods [...] may be largely attributed to the large-scale training data”* and that *“the more interesting application of FMs is for fine tuning on datasets with scarce/few labels.”* While do not disagree with this, we believe evaluating in the limited data regime is *not* critical for our study because we chose benchmarks from among those either explicitly proposed by the relevant community for evaluating foundation models (genomics and satellite) or broadly used to do so by the vast majority of its FMs (time series). If the main utility of FMs in a specific domain is in the few-shot regime then its *existing* benchmarks and evaluations should reflect this (developing new ones is out-of-scope of our paper).
>
> Still, to give a sense of the effect of different amounts of data, we pick the same subset of tasks and FMs as above and give each method only 10% of the original data (in time series we use 20% because the ETTh1 sequence is too short to use 10% while keeping the original context window). As shown in the tables below, we find even in these subsampled regimes that the trend we identified—that our baselines are competitive with leading FMs—remains consistent.
> #### Avg. genomics performance (↑) across 4 tasks: enhancers, H3, promoter_all, and splice_sites_all
> | Method  | 100% data | 10% data |
> |---|---|---|
> | HyenaDNA-1K  | 0.808  | 0.636 |
> | NT-Multispecies (500M) | 0.822 | 0.769 |
> | DASHA | 0.814 | 0.720 |
> #### Avg. satellite performance (↑) across 3 tasks: m-brickkiln, m-bigearthnet, and EuroSAT
> | Method  | 100% data | 10% data |
> |---|---|---|
> | CROMA-Large  | 90.45 | 83.76  |
> | SatMAE-Large | 90.43 | 80.75 |
> | DASHA | 90.24  | 83.04  |
> #### Avg. time series performance (↓) across 4 horizons of 2 tasks: ETTh1 and Weather
> | Method | 100% data | 20% data |
> |---|---|---|
> | GPT4TS | 0.332 | 0.349 |
> | MOMENT | 0.323 | 0.366 |
> | Auto-AR | 0.320 | 0.324 |

---

### Comment · Area_Chair_WTwP · 2024-11-22
**Discussion**

Dear reviewers,

The authors have responded to your reviews.

Until November 26th @ 2359 (AOE time) reviewers and authors can freely exchange responses, so if there any clarifications you require from the authors, now is the time to seek them!

Best,

AC

---

### Meta-Review · Area_Chair_WTwP · 2024-12-12

**Metareview:**

In this paper, the authors consider three modalities and compare "foundational model" performance to a standard supervised learning workflow.  They show that the latter can beat the former in all modalities considered. This is an interesting piece of work that sanity checks the "foundational model" paradigm. Reviewers appreciated the topic, and the rigorous experimental setup. There were criticisms, but the authors responded well to these which saw three reviewer scores raised. As this paper has entirely positive reviews, and no major flaws I believe this should be accepted.

**Additional Comments On Reviewer Discussion:**

There were good interactions between the authors and reviewers with the authors succeeding in addressing many of the concerns of the reviewers, and providing additional experiments as per their requests (including low-data tasks, and measures of computational cost). Three reviewers increased their scores (5,5,5 --> 6,6,8) giving this paper entirely positive reviews.

---

### Decision · Program_Chairs · 2025-01-22

Accept (Poster)